# The bacterial quorum sensing signal 2'-aminoacetophenone rewires immune cell bioenergetics through the Ppargc1a/Esrra axis to mediate tolerance to infection

**Arijit Chakraborty[1,2,3†], Arunava Bandyopadhaya[1,2†‡], Vijay K Singh[1,2], Filip Kovacic[1,3,4], Sujin Cha[1], William M Oldham[5], A Aria Tzika[1,2], Laurence G Rahme[1,2,3*]**

[1]Department of Surgery, Massachusetts General Hospital, and Harvard Medical School, Boston, United States; [2]Shriners Hospitals for Children Boston, Boston, United States; [3]Department of Microbiology, Harvard Medical School, Boston, United States; [4]Institute of Molecular Enzyme Technology, Heinrich Heine University Düsseldorf, Jülich, Germany; [5]Department of Medicine, Brigham and Women's Hospital and Harvard Medical School, Boston, United States

*For correspondence:
rahme@molbio.mgh.harvard.edu

†These authors contributed equally to this work

Present address: ‡Astellas Pharma Inc, Northbrook, United States

**Abstract** How bacterial pathogens exploit host metabolism to promote immune tolerance and persist in infected hosts remains elusive. To achieve this, we show that *Pseudomonas aeruginosa* (*PA*), a recalcitrant pathogen, utilizes the quorum sensing (QS) signal 2'-aminoacetophenone (2-AA). Here, we unveil how 2-AA-driven immune tolerization causes distinct metabolic perturbations in murine macrophages' mitochondrial respiration and bioenergetics. We present evidence indicating that these effects stem from decreased pyruvate transport into mitochondria. This reduction is attributed to decreased expression of the mitochondrial pyruvate carrier (*Mpc1*), which is mediated by diminished expression and nuclear presence of its transcriptional regulator, estrogen-related nuclear receptor alpha (Esrra). Consequently, Esrra exhibits weakened binding to the *Mpc1* promoter. This outcome arises from the impaired interaction between Esrra and the peroxisome proliferator-activated receptor gamma coactivator 1-alpha (Ppargc1a). Ultimately, this cascade results in diminished pyruvate influx into mitochondria and, consequently reduced ATP production in tolerized murine and human macrophages. Exogenously added ATP in infected macrophages restores the transcript levels of *Mpc1* and *Esrra and* enhances cytokine production and intracellular bacterial clearance. Consistent with the in vitro findings, murine infection studies corroborate the 2-AA-mediated long-lasting decrease in ATP and acetyl-CoA and its association with *PA* persistence, further supporting this QS signaling molecule as the culprit of the host bioenergetic alterations and *PA* persistence. These findings unveil 2-AA as a modulator of cellular immunometabolism and reveal an unprecedented mechanism of host tolerance to infection involving the Ppargc1a/Esrra axis in its influence on Mpc1/OXPHOS-dependent energy production and *PA* clearance. These paradigmatic findings pave the way for developing treatments to bolster host resilience to pathogen-induced damage. Given that QS is a common characteristic of prokaryotes, it is likely that 2-AA-like molecules with similar functions may be present in other pathogens.

### eLife assessment

This **important** study demonstrates that the *Pseudomonas aeruginosa*-derived quorum sensing signal, 2-aminoacetophenone, induces immune tolerization in macrophages by perturbing metabolism, particularly in the context of mitochondrial respiration and bioenergetics. The authors present **convincing** evidence for 2-aminoacetophenone-mediated reduction of pyruvate transport into mitochondria, with downstream effects that result in reduced ATP production in tolerized macrophages. The work will be of interest to those studying host-pathogen interactions.

## Introduction

Host tolerance is a fundamental mechanism of innate immunity. Studies have shown that innate immune cells following infection or exposure to microbial products may enter a state of immune tolerance characterized by reduced responsiveness toward microbial re-exposure a few hours later (*Foster et al., 2007*; *Bagchi et al., 2007*; *Masyutina et al., 2023*). Epigenetic and signaling-based mechanisms have been proposed to be involved in host immune tolerance (*Foster et al., 2007*; *Bagchi et al., 2007*; *Masyutina et al., 2023*). In recent years, several reports have highlighted the complex interplay between metabolic reprogramming and immunity (*O'Neill et al., 2016*; *Chi, 2022*). It was suggested that specific metabolic programs are activated in monocytes and macrophages upon exposure to infection and microbial products (*Tannahill et al., 2013*; *Galli and Saleh, 2020*). However, in host tolerance, the molecular mechanisms underlying immunometabolic reprogramming mediated by bacterial pathogens remain poorly understood.

Metabolic pathways are essential for generating energy for various cellular functions, including those performed by immune cells (*Wang et al., 2021*; *Karan et al., 2022*). Macrophages use metabolic pathways to generate energy and metabolites to adapt to changing environments and stimuli, thereby enabling them to cope with the needs of a fluctuating immune response (*Galli and Saleh, 2020*; *Bird, 2019*). Thus, properly functioning metabolic pathways are vital for immune cells to counteract pathogens. For instance, the crucial energy-carrying molecule adenosine triphosphate (ATP) is required to phagocytose pathogen-derived molecules efficiently.

*Pseudomonas aeruginosa (PA)*, a recalcitrant ESKAPE (*Enterococcus faecium*, *Staphylococcus aureus*, *Klebsiella pneumoniae*, *Acinetobacter baumannii*, *PA*, and *Enterobacter* sp.) pathogen that causes acute and persistent infections, secretes virulence-associated low molecular weight signaling molecules, several of which are regulated by quorum sensing (QS) (*Déziel et al., 2005*; *Xiao et al., 2006*; *Eickhoff and Bassler, 2018*; *Fuqua and Greenberg, 2002*) and able to modulate host immune responses (*Kariminik et al., 2017*; *Liu et al., 2015*; *Bandyopadhaya et al., 2012*; *Bandyopadhaya et al., 2016b*). QS is a cell density-dependent signaling system bacteria used to synchronize their activities (*Déziel et al., 2005*; *Xiao et al., 2006*; *Eickhoff and Bassler, 2018*; *Fuqua and Greenberg, 2002*). MvfR (a.k.a. PqsR), a critical QS transcription factor of *PA*, regulates the synthesis of many small molecules, including signaling molecules such as 2'-aminoacetophenone (2-AA) (*Déziel et al., 2005*; *Cao et al., 2001*; *Déziel et al., 2004*; *Kesarwani et al., 2011*). In vivo studies have demonstrated that 2-AA enables *PA* to persist in infected murine tissues by promoting innate immune tolerance through histone deacetylase 1 (HDAC1)-mediated epigenetic reprogramming (*Bandyopadhaya et al., 2012*; *Bandyopadhaya et al., 2016b*). 2-AA tolerization reprograms the host inflammatory signaling cascade by maintaining chromatin in a 'silent' state through increased HDAC1 expression and activity and decreased histone acetyltransferase (HAT) activity (*Bandyopadhaya et al., 2016b*). These changes also impact the protein-protein interaction between HAT and cyclic AMP response element-binding protein/HDAC1 and p50/p65 nuclear factor-κB subunits (*Bandyopadhaya et al., 2016b*; *Bandyopadhaya et al., 2017*). The 2-AA-mediated tolerization permits *PA* to persist in infected tissues (*Bandyopadhaya et al., 2012*; *Bandyopadhaya et al., 2016b*), underscoring the difference from lipopolysaccharide (LPS)-mediated tolerization, which instead leads to bacterial clearance and involves different HDACs (*Wheeler et al., 2008*).

Mitochondria, the 'powerhouse' of the cells, are crucial for the regulation, differentiation, and survival of macrophages and other immune cells (*Wang et al., 2021*). Our group's previous in vivo and in vitro studies indicated that 2-AA affects metabolic functions in skeletal muscle, which contains a high concentration of mitochondria (*Tzika et al., 2013*; *Bandyopadhaya et al., 2016a*; *Chakraborty et al., 2023*). Injection of 2-AA in murine skeletal muscle dampens the expression of

genes associated with OXPHOS and the master regulator of mitochondrial biogenesis peroxisome proliferator-activated receptor-γ coactivator-1 beta (*Ppargc1b*) (*Tzika et al., 2013*). Another *PA* QS molecule, 3-oxo-C12-HSL, attenuates the expression of *Ppargc1a* (*Maurice et al., 2019*). *Ppargc1a* and *Ppargc1b* belong to the PPARγ family of inducible transcriptional coactivators and are known regulators of mitochondrial metabolism (*Lelliott and Vidal-Puig, 2009*). The coactivator Ppargc1 proteins are involved in various cellular energy metabolic processes (*Supruniuk et al., 2017*; *Coppi et al., 2021*), including but not limited to mitochondrial metabolism (*Lin et al., 2005*). Recent studies have also emphasized the role of estrogen-related nuclear receptor α (*Esrra*) in coordinating metabolic capacity with energy demand in health and disease (*Huss et al., 2015*). Ppargc1a activates Esrra transcriptional activity through protein-protein interaction, which enhances the expression of *Esrra* (*Schreiber et al., 2003*; *Laganière et al., 2004*) and other mitochondrial genes involved in lipid metabolism and OXPHOS (*Villena, 2015*). Ppargc1a/Esrra axis has been extensively studied in cancer and established as a central regulatory node of energy metabolism that induces the global expression of genes involved in mitochondrial biogenesis and functions (*Ranhotra, 2010*; *Deblois and Giguère, 2011*; *Ranhotra, 2012*). *Esrra* has been shown to occupy the promoter regions of many genes participating in the TCA cycle and OXPHOS (*Charest-Marcotte et al., 2010*; *Eichner and Giguère, 2011*), including the mitochondrial pyruvate carrier (*Mpc1*). In human renal carcinoma cells, Ppargc1a/Esrra interaction results in efficient activation of *Mpc1* expression and transport of pyruvate into mitochondrion for efficient OXPHOS and energy production (*Koh et al., 2018*).

Given that our previous studies pointed to the 2-AA effect on mitochondrial functions (*Tzika et al., 2013*; *Bandyopadhaya et al., 2016a*; *Chakraborty et al., 2023*) and that mitochondria are crucial for the regulation, differentiation, and survival of macrophages and other immune cells (*Wang et al., 2021*), we investigated how 2-AA may mediate cellular metabolic reprogramming in tolerized immune cells by examining the link between the Ppargc1a/Esrra axis and 2-AA-mediated immune tolerance. Our findings uncovered the crucial action of 2-AA on energy homeostasis and metabolism in tolerized immune cells through perturbances of the Esrra/Ppargc1a interaction, Mpc1-mediated pyruvate transport, and the production of the key energy metabolism molecules, ATP, and acetyl-CoA and their association with the persistence of *PA* in mammalian tissues.

## Results

### Tolerization by 2-AA impacts the generation of crucial energy metabolites in macrophages

Following infection or exposure to microbial products, innate immune cells may enter a state of tolerance several hours later, as depicted by their diminished responsiveness or unresponsiveness to microbial re-exposure (*Foster et al., 2007*; *Bagchi et al., 2007*; *Masyutina et al., 2023*; *Divangahi et al., 2021*). Given that 2-AA-tolerized cells are non-responsive to 2-AA re-exposure, we investigated the impact of 2-AA on energy homeostasis and metabolism in non-tolerized and tolerized BMDM cells (*Figure 1*). We quantified the levels of ATP and acetyl-CoA, which is a fuel for ATP production, in a series of in vitro experiments in which murine BMDM cells were exposed to 2-AA (first 2-AA exposure) or re-exposed (second 2-AA exposure) to respectively determine their activation, and tolerance as well as memory to this QS molecule (*Figure 1A*). Initially, following the first 2-AA exposure (black bars), a significant increase in the concentrations of intracellular ATP (*Figure 1B*) and acetyl-CoA (*Figure 1C*) was observed in BMDM cells exposed to 2-AA for 1 and 6 hr compared to their corresponding naïve control cells (gray bars). However, cells exposed to 2-AA for 48 hr (black bar) appeared to have entered a state of tolerance as ATP and acetyl-CoA levels were decreased compared to 2-AA exposures for a shorter time (*Figure 1B and C*). The tolerance of these cells as well as their memory to 2-AA was determined by their responsiveness to a second 2-AA exposure (*Figure 1A*). To achieve this, these cells were washed to remove 2-AA and allowed to rest for 24 or 106 hr in the absence of 2-AA before receiving the second 2-AA exposure for 1 or 6 hr (*Figure 1A*). *Figure 1B and C* (red bars) shows the unresponsiveness of BMDM cells to 2-AA re-exposure as evidenced by the similar levels of ATP and acetyl-CoA compared to cells exposed to 2-AA for 48 hr (first exposure black bars). The sustained unresponsiveness of the cells to re-exposure indicates that the cells exhibited innate immunological memory, which could not be reverted by 2-AA re-exposure. Similar responsiveness patterns of ATP and acetyl-CoA as with BMDM cells were observed in murine macrophage RAW 264.7 cells

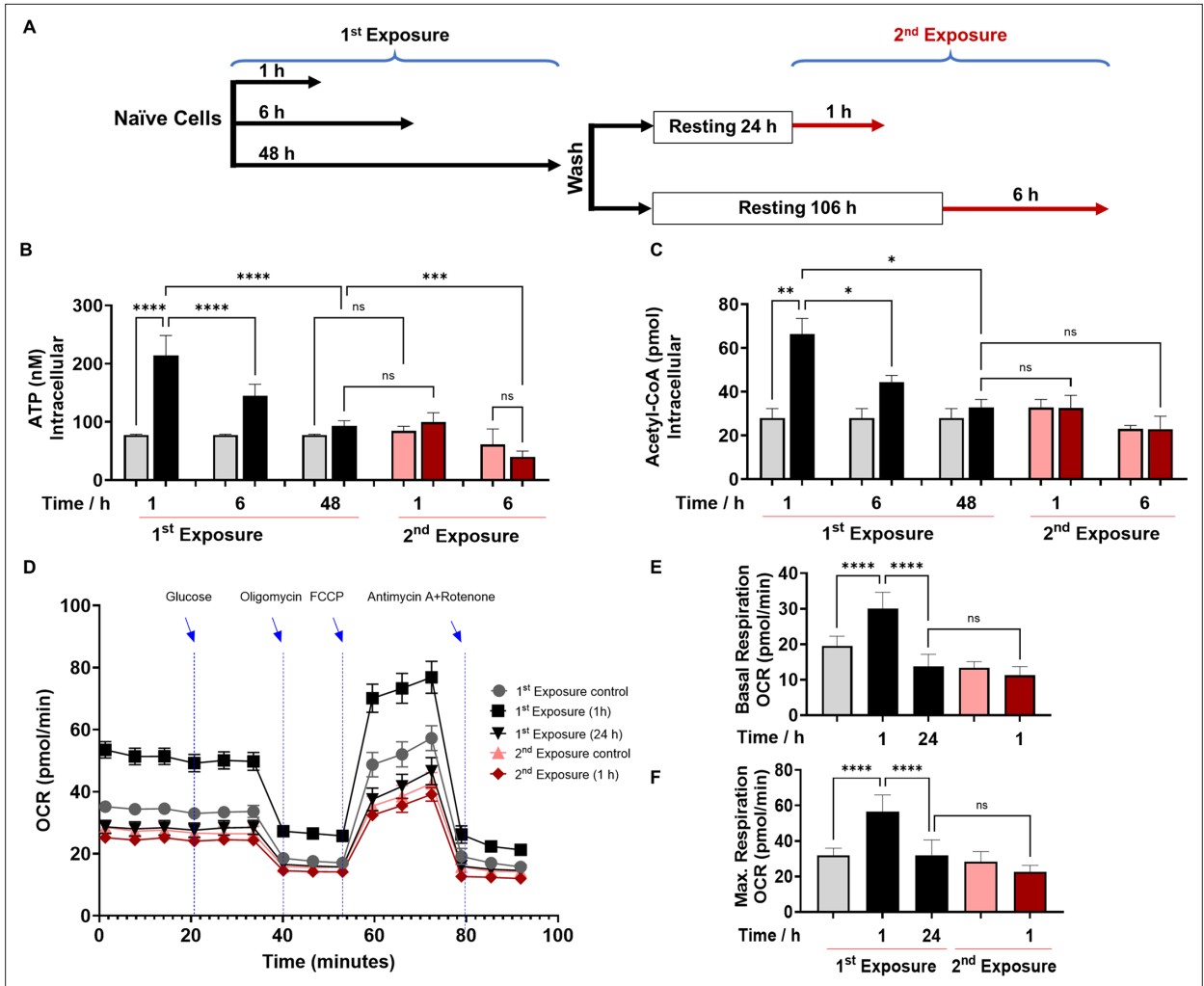

**Figure 1.** 2′-Aminoacetophenone (2-AA) tolerization decreases crucial metabolites of cellular energy and affects mitochondrial respiration in mouse BMDM. (**A**) Schematic representation showing experimental design: naïve cells were exposed to 2-AA for 1, 6, or 48 hr (black). Cells exposed to 2-AA (400 µM) for 48 hr were washed, rested for 24 or 106 hr, and re-exposed (200 µM) for 1 or 6 hr (red), respectively. The same color code, black cells after first exposure and red cells after second exposure, was kept throughout the manuscript with corresponding controls in gray and pink, respectively. The levels of (**B**) adenosine triphosphate (ATP) and (**C**) acetyl-CoA in BMDM cells after first and second 2-AA exposure. (**D**) Real-time oxygen consumption rate (OCR) traces were recorded using a Seahorse XF analyzer and normalized to protein content. Cells were exposed to 2-AA for 1 or 24 hr (black), washed and rested for 24 hr, and re-exposed for 1 hr (red). Mitochondrial respiratory parameters, (**E**) basal respiration, and (**F**) maximal respiration. Data are presented as mean ± SD, n≥4, *p<0.05, **p<0.01, ***p<0.001, and ns indicates no significant difference. One-way ANOVA followed by Tukey's post hoc test was applied.

The online version of this article includes the following source data and figure supplement(s) for figure 1:

**Source data 1.** The numerical data used to generate *Figure 1B*.

**Source data 2.** The numerical data used to generate *Figure 1C*.

**Source data 3.** The numerical data used to generate *Figure 1D*.

**Figure supplement 1.** 2′-Aminoacetophenone (2-AA) tolerization decreases metabolites in murine RAW 264.7 (**A–B**) and human THP-1 (**C–D**) cells.

**Figure supplement 1—source data 1.** The numerical data used to generate *Figure 1—figure supplement 1*.

**Figure supplement 2.** Adenosine triphosphate (ATP) levels in murine BMDM (**A**) cells with and without 2′-aminoacetophenone (2-AA) (400 µM) or lipopolysaccharide (LPS) stimulation (100 µg/mL).

**Figure supplement 2—source data 1.** The numerical data used to generate *Figure 1—figure supplement 2*.

**Figure supplement 3.** Effect of 2′-aminoacetophenone (2-AA) on mitochondrial spare respiratory capacity in BMDM non-tolerized (black) and tolerized (red) macrophages.

**Figure supplement 3—source data 1.** The numerical data used to generate *Figure 1—figure supplement 3*.

*Figure 1 continued on next page*

*Figure 1 continued*

**Figure supplement 4.** Transmission electron microscopy (TEM) images showing structural alterations of mitochondria in 2'-aminoacetophenone (2-AA) exposed macrophages for 48 hr.

(*Figure 1—figure supplement 1A* and *Figure 1—figure supplement 1B*) or human monocyte THP-1 cells (*Figure 1—figure supplement 1C* and *Figure 1—figure supplement 1D*) following exposure or re-exposure to 2-AA. These findings further support our observations with BMDM cells and confirm that either cell type can be used to study responses to 2-AA. These results reinforce the notion that 2-AA impacts crucial energy metabolites in macrophages.

Moreover, we sought to determine whether 2-AA-tolerized BMDM cells would respond to heterologous stimulus. To test this, we re-exposed the cells to LPS, an outer membrane bacterial component that strongly induces macrophages. LPS re-exposure of the 2-AA-tolerized BMDM cells did not significantly (p=0.9) increase ATP levels (*Figure 1—figure supplement 2*), indicating that 2-AA cross-tolerized the cells to this heterologous bacterial immunostimulant. On the other hand, as expected, we observed increased ATP levels following LPS stimulation of the non-tolerized BMDM cells (*Figure 1—figure supplement 2*). The broad spectrum of 2-AA cross-tolerization strongly suggests an interplay of epigenetic mechanism rather than ligand-receptor-mediated signaling-based mechanism in bringing out the observed 2-AA-mediated tolerance.

## Tolerization by 2-AA leads to a quiescent state by reducing cell bioenergetics and compromising OXPHOS

To further investigate the 2-AA's impact on the bioenergetics of macrophages, we assessed by Seahorse assay the oxygen consumption rate (OCR) as an index of OXPHOS in 2-AA exposed (for 1 and 24 hr) and re-exposed (second exposure for 1 hr) BMDM cells (*Figure 1D*). Values of basal (*Figure 1E*) and maximal (*Figure 1F*) mitochondrial respiration and spare respiratory capacity (*Figure 1—figure supplement 3*) interpreted from these measurements revealed that 1 hr 2-AA exposure of naïve cells significantly increased OCR compared to naïve control cells. At 24 hr post-2-AA exposure, cells had a significantly reduced basal OCR level than the control BMDM cells, indicating decreased OXPHOS and an overall quiescent phenotype of tolerized macrophages. Re-exposure of tolerized macrophages with 2-AA also did not augment maximal and basal OCR (*Figure 1D–F*), indicating the unresponsiveness of tolerized BMDM cells. Ultramicroscopic examination of RAW 264.7 cells exposed to 2-AA for 48 hr although shows the same number of mitochondria as in control cells, their mitochondrial morphology is altered, appearing smaller and round and having reduced cristae indicating dysfunctional mitochondria (*Figure 1—figure supplement 4*).

These OCR data are consistent with the previous direct measurements of ATP (*Figure 1B*) and acetyl-CoA (*Figure 1C*) and support the notion that 2-AA tolerization induces a quiescent phenotype in these cells characterized by defective OXPHOS. These findings confirm that 2-AA mediates energy homeostatic and metabolic alterations in the immune cells and that 2-AA tolerization arrests these cells in a sustained 2-AA-unresponsive state.

## The pyruvate transport into mitochondria is decreased in tolerized macrophages

Since acetyl-CoA and ATP levels were reduced due to 2-AA tolerization, we investigated whether the mitochondrial dysfunction observed is related to pyruvate metabolism. Pyruvate links glycolysis with mitochondrial production of acetyl-CoA and ATP via the tricarboxylic acid (TCA) cycle and OXPHOS (*Galli and Saleh, 2020*). RAW 264.7 cells initially exposed to 2-AA for 1 or 3 hr (black bars) exhibited significantly higher concentrations of cytosolic and mitochondrial pyruvate compared to the corresponding naïve control cells (*Figure 2A*). Pyruvate significantly decreased over time in cytosolic and mitochondrial fractions as cells were entering into the tolerized state (*Figure 2A*). Tolerized cells remain unresponsive to 2-AA re-exposure (red bars), exhibiting no significant change in the concentration of pyruvate in the cytosolic or mitochondrial fraction (*Figure 2A*). A similar result was observed in THP-1 cells (*Figure 2—figure supplement 1A* and *Figure 2—figure supplement 1B*).

To gain a better understanding of the 2-AA-mediated reductions in mitochondrial pyruvate, ATP production, and acetyl-CoA levels, we sought to determine the involvement of the Mpc1 (*Koh et al.,*

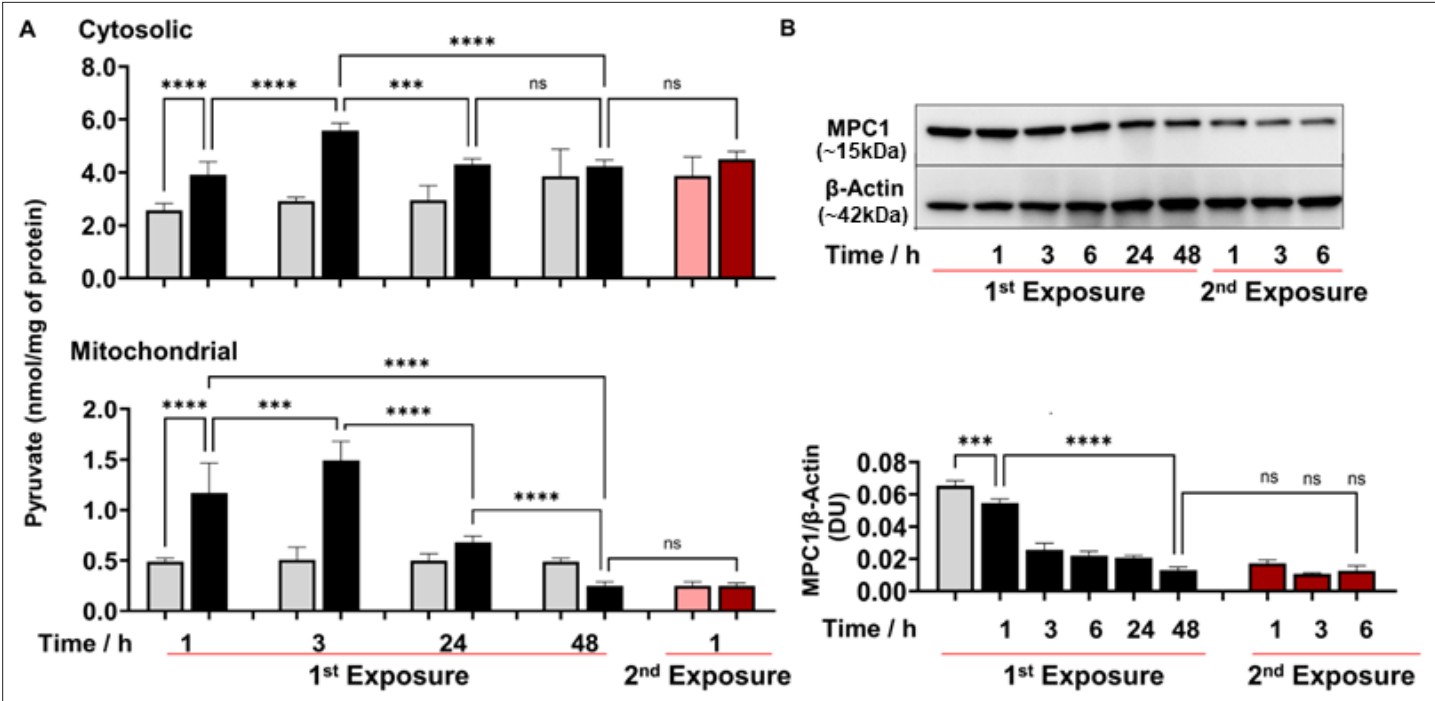

**Figure 2.** 2'-Aminoacetophenone (2-AA) perturbs the mitochondrial Mpc1-mediated import and metabolism of pyruvate. (**A**) Cytosolic and mitochondrial pyruvate levels following 2-AA exposure (black) or re-exposure (red) and corresponding controls in gray and pink, respectively, for indicated time points. (**B**) Representative western blot and results of densitometric analysis of Mpc1 protein levels following 2-AA exposure or re-exposure for indicated time points. β-Actin was used as a control. Corresponding controls are shown in gray or pink, respectively. Mean ± SD is shown, n=3, ***p<0.001, ****p<0.0001, and ns indicates no significant difference. One-way ANOVA followed by Tukey's post hoc test was applied.

The online version of this article includes the following source data and figure supplement(s) for figure 2:

**Source data 1.** The numerical data used to generate *Figure 2A*.

**Source data 2.** Uncropped and labeled blots for *Figure 2B*.

**Source data 3.** Raw unedited blots for *Figure 2B*.

**Figure supplement 1.** 2'-Aminoacetophenone (2-AA) tolerization decreases pyruvate levels in human THP-1 cells.

**Figure supplement 1—source data 1.** The numerical data used to generate *Figure 2—figure supplement 1A*.

**Figure supplement 1—source data 2.** The numerical data used to generate *Figure 2—figure supplement 1B*.

**Figure supplement 1—source data 3.** Uncropped and labeled electron micrographs for *Figure 2—figure supplement 1*.

**Figure supplement 1—source data 4.** Raw unedited electron micrographs for *Figure 2—figure supplement 1*.

*2018*) that is crucial in transporting pyruvate from the cytosol into the mitochondria (*Xue et al., 2021*). Western blot studies of whole-cell lysates of RAW 246.7 cells exposed to 2-AA showed a significant decrease over time in the abundance of Mpc1 protein, and tolerized cells remain unresponsive to 2-AA re-exposure (*Figure 2B*). Together, these findings suggest that 2-AA tolerization dysregulates Mpc1-mediated transport of the pyruvate into mitochondria.

## 2-AA tolerization suppresses Esrra binding to the *Mpc1* promoter and the interaction between Ppargc1a and Esrra

Given that the *Ppargc1a/Esrra* axis is the transcriptional regulatory node that activates the expression of *Mpc1* (*Schreiber et al., 2003*), we determined the protein levels of Esrra. Western blot studies of whole-cell lysates showed a significant decrease over time in the abundance of Esrra protein following first exposure of RAW 246.7 cells to 2-AA, while tolerized cells remain unresponsive to 2-AA re-exposure (*Figure 3A*). By assessing the cytosolic and nuclear abundance of Esrra in tolerized cells, we found that both fractions had lower levels of Esrra protein than their corresponding control cells (*Figure 3B*).

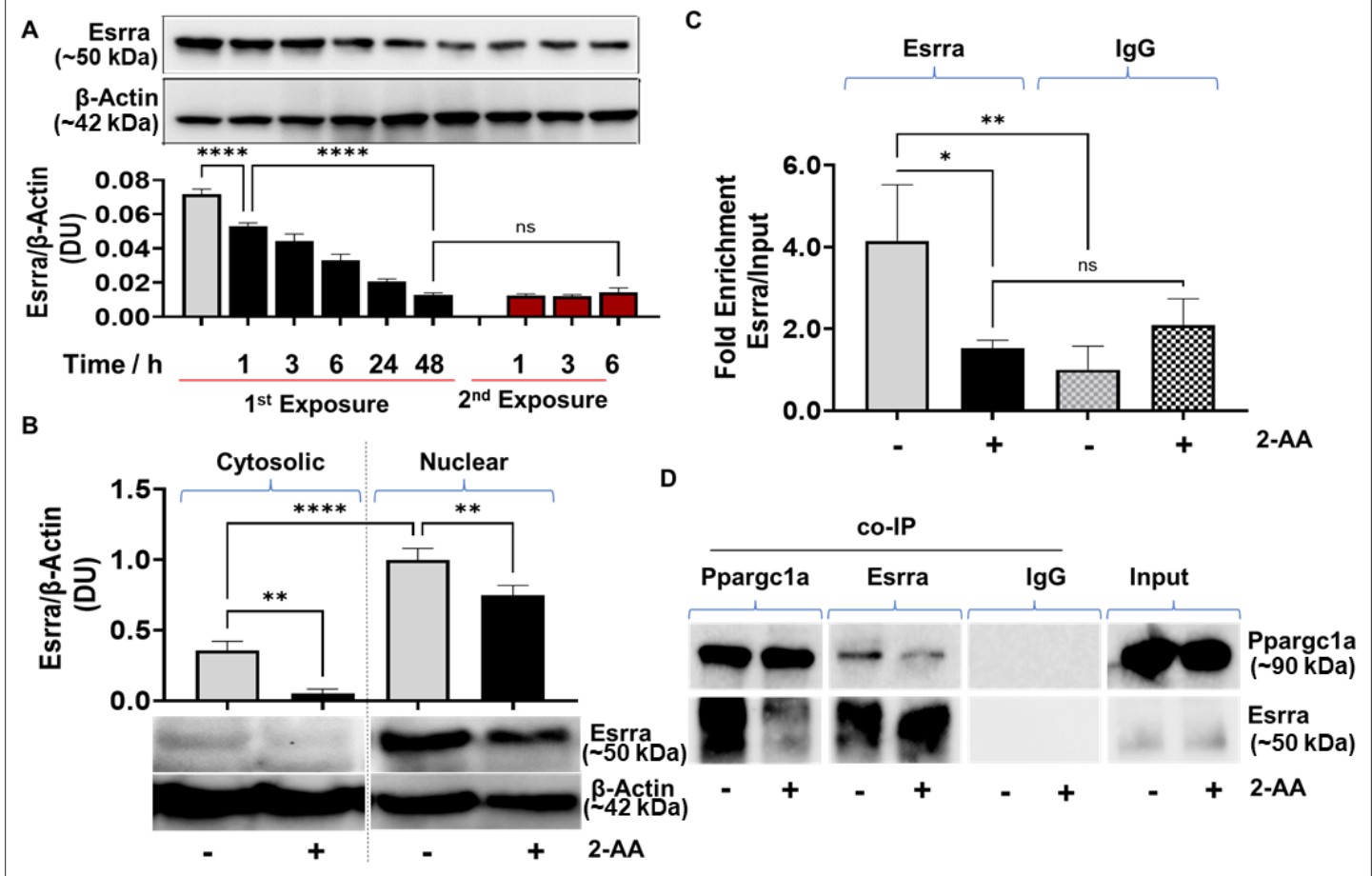

**Figure 3.** 2'-Aminoacetophenone (2-AA)-mediated macrophage tolerization deranges Ppargc1a/Esrra-dependent metabolic programming.
(**A**) Representative western blot and results of densitometric analysis of Esrra protein levels following 2-AA exposure or re-exposure for indicated time points. β-Actin was used as a control. (**B**) Western blots and its corresponding densitometric analysis of Esrra in cytoplasmic or nuclear lysates of tolerized macrophages exposed to 2-AA for 24 hr or not exposed to 2-AA. (**C**) Chromatin immunoprecipitation (ChIP)-qPCR assay of Esrra binding at the *Mpc1* promoter in RAW 264.7 tolerized macrophages exposed to 2-AA (200 µM) for 24 hr (black) compared to untreated control macrophages (gray). IgG served as a negative control. (**D**) Representative western blot of co-immunoprecipitation (co-IP) studies of Esrra and Ppargc1a in nuclear extracts of 2-AA-tolerized (24 hr) and control RAW 264.7 cells. Pull-down with IgG served as a negative control. 2-AA-tolerized macrophages are shown in black, and untreated control macrophages in gray. Mean ± SD is shown, n ≥3, *p<0.05, **p<0.01, ****p<0.0001, and ns indicates no significant difference. One-way ANOVA followed by Tukey's post hoc test was applied.

The online version of this article includes the following source data for figure 3:

**Source data 1.** The numerical data used to generate *Figure 3B*.

**Source data 2.** The numerical data used to generate *Figure 3C*.

**Source data 3.** Uncropped and labeled blots for *Figure 3A*.

**Source data 4.** Raw unedited blots for *Figure 3A*.

**Source data 5.** Uncropped and labeled blots for *Figure 3B*.

**Source data 6.** Raw unedited blots for *Figure 3B*.

**Source data 7.** Uncropped and labeled blots for *Figure 3D*.

**Source data 8.** Raw unedited blots for *Figure 3D*.

Furthermore, because an Esrra-binding putative DNA motif (TNAAGGTCA) has been identified upstream of the *Mpc1* promoter in humans (*Koh et al., 2018*), and we found the same motif to be present 1.5 kB upstream of the transcription start site of mouse *Mpc1* promoter, we analyzed if tolerization led to reduced Esrra binding to the *Mpc1* promoter motif TNAAGGTCA. Chromatin immunoprecipitation (ChIP) assay followed by qPCR analysis revealed that in tolerized cells, Esrra binds

approximately fourfold less efficiently to the putative binding site on the *Mpc1* than in corresponding control cells (*Figure 3C*).

Since the transcriptional activity of Esrra is regulated through a protein-protein interaction with the transcriptional coactivator Ppargc1a (*Schreiber et al., 2003*), we assessed whether the formation of the Ppargc1a/Esrra complex is affected in 2-AA-tolerized cells. Using paraformaldehyde fixed nuclear lysates of RAW 264.7 tolerized and naïve cells, we performed a co-immunoprecipitation (co-IP) of Ppargc1a and reversed co-IP with Esrra followed by immunoblotting (*Figure 3D*). As shown in *Figure 3D* less Esrra was detected when Ppargc1a was pulled down in the presence of 2-AA than in naïve control cells. Moreover, reverse co-IP with Esrra protein showed lower levels of Ppargc1a in the presence of 2-AA than in naïve control cells (*Figure 3D*). Control IgG showed no detectable levels of Esrra following nuclear fraction pull-down. These results indicate that in the tolerized macrophages, Ppargc1a/Esrra interaction is impaired (*Figure 3D*).

Moreover, since Ppargc1a enhances the transcriptional activity of Esrra through their interaction, and *Esrra* transcription is regulated via an autoregulatory loop (*Laganière et al., 2004*), we

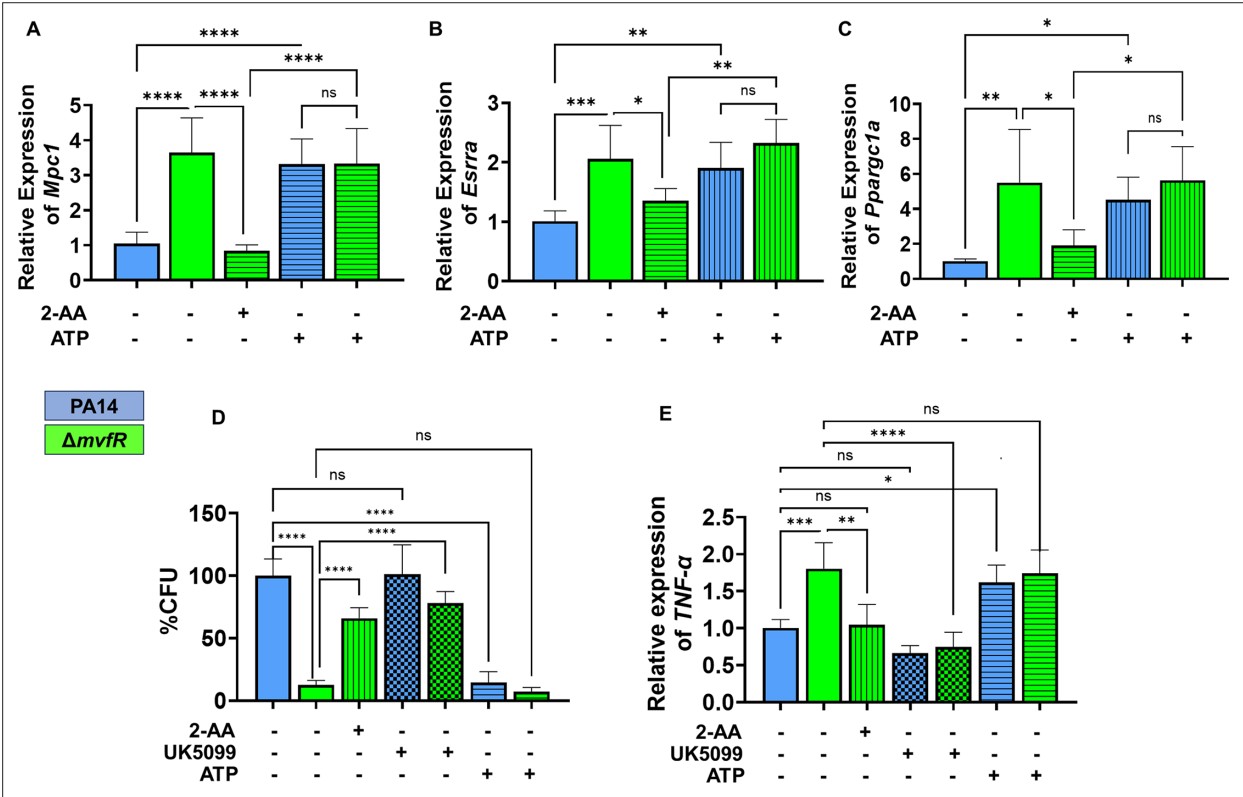

**Figure 4.** Increased intracellular burden in macrophages is associated with decreased expression of *Mpc1, Esrra,* and *TNF-α genes*. Real-time PCR analysis of *Mpc1* (**A**), *Esrra* (**B**), and *Ppargc1a* (**C**) expression in RAW 246.7 macrophages infected with PA14 or *ΔmvfR* in the presence or absence of exogenous addition of 2'-aminoacetophenone (2-AA) or adenosine triphosphate (ATP) for 6 hr as indicated. Transcript levels were normalized to 18S-rRNA. PA14-infected cells served as controls. (**D**) The intracellular burden of PA14 or *ΔmvfR* of infected macrophages in the presence or absence of exogenous addition of 2-AA, UK5099, or ATP. Untreated cells infected with PA14 were set as 100%. (**E**) Real-time PCR analysis of TNF-α expression in RAW 246.7 macrophages infected with PA14 or *ΔmvfR* in the presence or absence of exogenous addition of 2-AA, ATP, or UK5099. Transcript levels were normalized to 18S-rRNA. PA14-infected cells served as controls. The compound concentration used for UK5099 was 10 μM and ATP 20 μM. Mean ± SD is shown, n=3, *p<0.05, **p<0.01, ***p<0.001, ****p<0.0001, and ns indicates no significant difference. One-way ANOVA followed by Tukey's post hoc test was applied.

The online version of this article includes the following source data for figure 4:

**Source data 1.** The numerical data used to generate *Figure 4A*.

**Source data 2.** The numerical data used to generate *Figure 4B*.

**Source data 3.** The numerical data used to generate *Figure 4C*.

**Source data 4.** The numerical data used to generate *Figure 4D*.

**Source data 5.** The numerical data used to generate *Figure 4E*.

examined the effect of tolerization on the transcription of *Ppargc1a*, *Esrra*, and its target gene *Mpc1*. Using RAW 264.7 cells infected with the wild-type *PA* strain PA14 or isogenic Δ*mvfR* mutant, which does not produce 2-AA (*Kesarwani et al., 2011*) we observed lower RNA transcript levels of *Mpc1* (*Figure 4A*), *Esrra* (*Figure 4B*), and *Ppargc1a* (*Figure 4C*) in PA14 compared to Δ*mvfR*. The addition of 2-AA to Δ*mvfR* decreased the levels of these gene transcripts to levels similar to the PA14 infection condition, supporting the role of 2-AA in effect. To test if the observed effects are due to Mpc1-dependent reduction of mitochondrial ATP generation, we supplemented the macrophages with ATP. The findings indicate that the addition of ATP in PA14-infected cells elevated the transcript levels of *Esrra*, *Mpc1*, and *Ppargc1a*, reaching levels similar to those observed in Δ*mvfR* infected macrophages (*Figure 4A, B, and C*). These results indicate that reduced Mpc1 function is due to 2-AA tolerization on the transcriptional activation of *Mpc1* through the *Ppargc1a*/*Esrra* axis.

## 2-AA tolerization impairs macrophage-mediated intracellular bacterial clearance through decrease in Mpc1-mediated pyruvate import, ATP, and TNF-α levels

2-AA mediates persistence of *PA* in vivo, dampens pro-inflammatory responses, and increases the intracellular burden of this pathogen in macrophages via epigenetic modifications (*Bandyopadhaya et al., 2012*; *Bandyopadhaya et al., 2016b*; *Chakraborty et al., 2023*). Here, we used RAW 264.7 cells infected with the PA14 or the 2-AA-deficient Δ*mvfR* mutant to assess the clearance of *PA* by macrophages. Macrophages infected with PA14 showed increased bacterial burden than the cells infected with Δ*mvfR*, and 2-AA addition to Δ*mvfR* led to increased bacterial burden (*Figure 4D*). Furthermore, we used UK5099 and ATP to interrogate whether Mpc1-mediated mitochondrial pyruvate import and bioenergetics are linked to the clearance of *PA* intracellular burden. The addition of the UK5099 inhibitor strongly enhanced the bacterial intracellular burden in Δ*mvfR* infected macrophages compared to the non-inhibited Δ*mvfR* infected cells, reaching a similar burden to those infected with PA14 (*Figure 4D*). Conversely, exogenously added ATP to macrophages infected with PA14 strongly reduced the *PA* intracellular burden (*Figure 4D*).

Given that 2-AA tolerization decreases the expression of pro-inflammatory cytokine TNF-α by hypoacetylating the core histone 3 lysine 18 acetylation (H3K18ac) mark at TNF-α promoter (*Bandyopadhaya et al., 2016b*), we investigated the link between bioenergetics and TNF-α expression in infected macrophages. UK5099 or ATP was added exogenously to infected macrophages with PA14 or Δ*mvfR* (*Figure 4E*). As shown in *Figure 4E*, PA14-infected cells showed lower TNF-α transcript levels compared to the Δ*mvfR* infected cells. Supplementation of 2-AA to Δ*mvfR* infected cells led to a decrease in the TNF-α transcript levels (*Figure 4E*). The addition of the Mpc1 inhibitor, UK5099, in Δ*mvfR* infected cells counteracted the increase in TNF-α transcript levels observed in Δ*mvfR* infected cells in the absence of UK5099 (*Figure 4E*). Conversely, exogenous addition of ATP to PA14-infected cells enhanced the transcription of TNF-α compared to the untreated PA14-infected cells, while no difference in TNF-α expression levels were observed in Δ*mvfR* infected macrophages in presence or absence of ATP (*Figure 4E*). These findings strongly suggest that 2-AA tolerization severely alters macrophages' ability to facilitate the clearance of *PA* intracellular burden, via the reduction in Mpc1-mediated pyruvate import into mitochondria, ATP levels, and TNF-α transcription.

## In vivo studies corroborate the 2-AA-mediated decrease in vitro of the central metabolic fuel acetyl-CoA and the energy-carrying molecule ATP and their association with *PA* persistence

To determine whether the decrease in the key metabolites mediated by 2-AA is also observed in vivo during infection, we quantified the levels of ATP and acetyl-CoA in murine spleen tissues at 1, 5, and 10 days (*Figure 5*). This organ was selected for our immunometabolic studies due to its role in regulating not only local but also systemic (whole-body) immune responses, facilitated by various immune cells including macrophages (*Bronte and Pittet, 2013*). Mice were infected with *PA* strain PA14, or isogenic mutant Δ*mvfR*, which does not produce 2-AA (*Kesarwani et al., 2011*). It is important to note that it is not possible to generate or utilize a bacterial mutant that is defective in 2-AA only because 2-AA is formed by spontaneous decarboxylation rather than by an enzyme-catalyzed reaction (*Starkey et al., 2014*; *Dulcey et al., 2013*; *Fetzner and Drees, 2013*). Therefore, animals that were either infected with Δ*mvfR* and received 2-AA (Δ*mvfR* + 2-AA) at the time of infection or uninfected

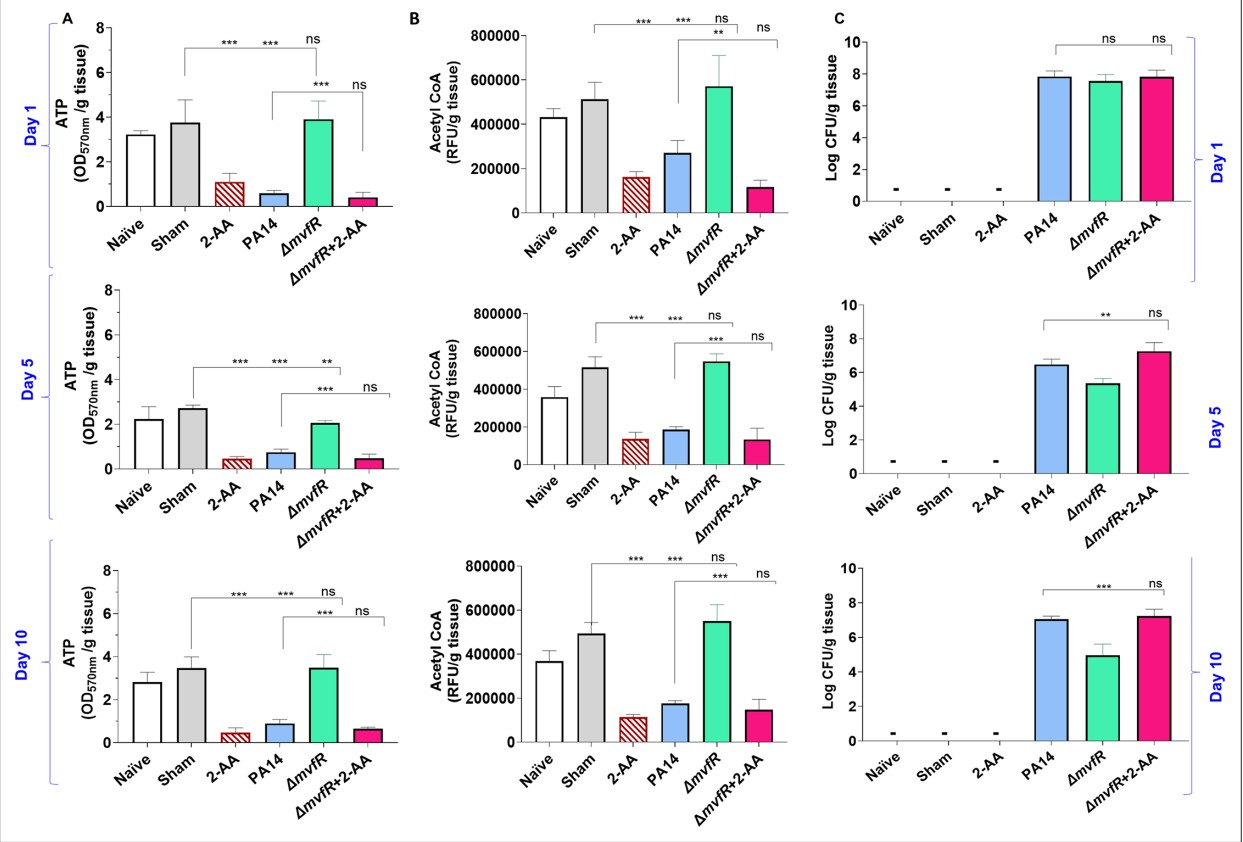

**Figure 5.** 2'-Aminoacetophenone (2-AA) promotes a long-lasting decrease in adenosine triphosphate (ATP), acetyl-CoA levels, and bacterial persistence in *P. aeruginosa* (*PA*)-infected mice. (**A**) ATP and (**B**) acetyl-CoA concentrations in the spleens of mice infected with *PA* wild-type (PA14), the isogenic mutant Δ*mvfR*, Δ*mvfR* injected with 2-AA at the time of infection (Δ*mvfR* + 2-AA), or non-infected but injected with 2-AA (6.75 mg/kg). (**C**) Bacterial burden in muscles expressed as colony-forming unit (CFU) count was analyzed using the Kruskal-Wallis non-parametric test with Dunn's post-test; ***p<0.001, and ns indicates no significant difference. Control mice groups: naïve were not given 2-AA; mice receiving 2-AA were given a single intraperitoneal injection of 2-AA; sham represents a burn/PBS group since the burn and infection model was used. Results of three independent replicates with four mice per group for 1, 5, and 10 days are shown. Means ± SD are shown, *p<0.05, **p<0.01, ***p<0.001, and ns indicate no significant difference. One-way ANOVA followed by Tukey's post hoc test was applied.

The online version of this article includes the following source data and figure supplement(s) for figure 5:

**Source data 1.** The numerical data used to generate *Figure 5A*.

**Source data 2.** The numerical data used to generate *Figure 5B*.

**Source data 3.** The numerical data used to generate *Figure 5C*.

**Figure supplement 1.** Exposure and re-exposure to 2'-aminoacetophenone (2-AA) promotes a long-lasting decrease in adenosine triphosphate (ATP) and acetyl-CoA levels and sustains bacterial presence in mice receiving first exposure to 2-AA by injecting 2-AA and second exposure through infection with PA14 or Δ*mvfR* 4 days post-2-AA injection.

**Figure supplement 1—source data 1.** The numerical data used to generate *Figure 5—figure supplement 1*.

mice injected with a single dose of 2-AA served as direct controls (*Figure 5*). Naïve and sham mice groups served as additional basal controls (*Figure 5*).

Infection with PA14 or uninfected mice injected with 2-AA led to a significant decrease in both ATP (*Figure 5A*) and acetyl-CoA (*Figure 5B*) concentrations in spleen tissues compared to Δ*mvfR* infected mice that sustained higher ATP and acetyl-CoA levels similar to naïve and sham control groups across the time points tested. However, in mice infected with Δ*mvfR* and 2-AA injected (Δ*mvfR* + 2-AA) at the time of infection, ATP and acetyl-CoA concentrations in the spleen decreased to levels comparable to those of PA14-infected animals, strongly indicating the biological function of 2-AA in decreasing these key metabolites. These in vivo findings further support the adverse action of 2-AA on host energy homeostasis and metabolism observed in in vitro studies.

The metabolic alterations observed are associated with the host tolerance to *PA* persistence (**Figure 5C**). Using *Pseudomonas* isolation agar plates, we evaluated the bacterial load at the infection site over the course of 10 days, obtaining samples at 1, 5, and 10 days. At 1 day post-infection, mice infected with PA14, Δ*mvfR,* or ΔmvfR + 2-AA exhibited no difference in bacterial burden (**Figure 5C**), verifying the ability of all strains to establish infection. At 5 and 10 days post-infection, mice infected with PA14 or ΔmvfR + 2-AA sustained the bacterial burden at a significantly higher bacterial burden over time compared to those infected with Δ*mvfR* (**Figure 5C**).

To strengthen the relevance of our in vivo data, we performed additional in vivo experiments. In this set of in vivo studies, mice received the first exposure to 2-AA by injecting 2-AA only and the second exposure through infection with PA14 or Δ*mvfR* 4 days post-2-AA injection. As shown in **Figure 5—figure supplement 1**, the levels of ATP and acetyl-CoA in the spleen of infected animals and the enumeration of the bacterial counts were similar between PA14 and Δ*mvfR* receiving the first 2-AA exposure and agree with the 'one-shot infection' findings presented in **Figure 5** with the PA14 or ΔmvfR + 2-AA infected mice or those receiving 2-AA only. These results are consistent with our previous findings, showing that 2-AA impedes the clearance of PA14 (**Bandyopadhaya et al., 2012**; **Bandyopadhaya et al., 2016b**) and provide compelling evidence that the metabolic alterations identified may favor *PA* persistence in infected tissues.

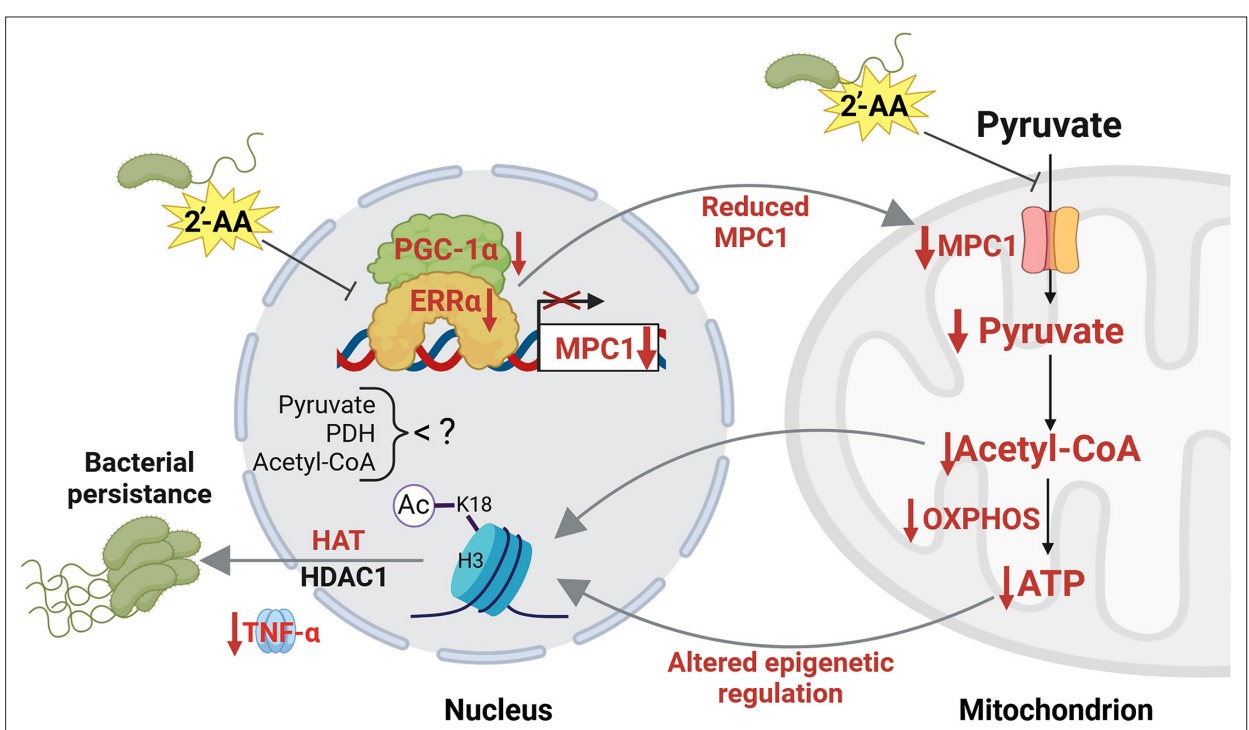

**Figure 6.** Proposed mechanism by which 2'-aminoacetophenone (2-AA) impairs bioenergetics through the inhibition of Mpc1-mediated pyruvate transport into mitochondria and its impact on the Ppargc1a/Esrra axis. 2-AA-tolerized macrophages exhibit diminished pyruvate levels in mitochondria due to the decreased expression of *Mpc1*, a consequence of the 2-AA impact on the interaction of Esrra with the transcriptional coactivator Ppargc1a for the effective transcription of *Esrra* since *Esrra* controls its own transcription and that of *Mpc1*. In the presence of 2-AA, the weakened interaction between Esrra and Ppargc1a results in reduced expression of *Mpc1* and *Esrra*. The reduction in mitochondrial pyruvate levels leads to decreased acetyl-CoA and adenosine triphosphate (ATP) levels, which modulate histone deacetylase 1 (HDAC1)- and histone acetyltransferase (HAT)-catalyzed remodeling of H3K18 acetylation. The diminished levels of this epigenetic mark have previously been associated with an increased intracellular presence of bacteria in macrophages, as demonstrated by our group (**Bandyopadhaya et al., 2016b**). Pathways, proteins, and metabolites that are negatively affected are indicated in red, while positively affected are denoted in black. Figure 6 was created with BioRender.com, and published using a CC BY-NC-ND license with permission.

## Discussion

The *PA* signaling molecule 2-AA that is abundantly produced and secreted in human tissues (*Kesarwani et al., 2011*; *Bandyopadhaya et al., 2017*) is the first QS molecule that epigenetically reprograms immune functions, promotes immune tolerance, and sustains *PA's* presence in host tissues (*Bandyopadhaya et al., 2012*; *Bandyopadhaya et al., 2016b*; *Bandyopadhaya et al., 2017*; *Chakraborty et al., 2023*). The present study provides insights into the mechanistic actions of 2-AA on cellular metabolism that contribute to host immune tolerance to *PA* persistence. We uncover that this signaling molecule causes distinct metabolic alterations in macrophages' mitochondrial respiration and energy production promoted via the Ppargc1a/Esrra axis and Mpc1-dependent OXPHOS that links the TCA cycle to the production of ATP and energy homeostasis (*Figure 6*). Our results show that 2-AA tolerization decreases ATP the crucial energy metabolite and the histone acetylation metabolite acetyl-CoA and in vivo, implicating the importance of the key energy-producing mitochondrial process in immune tolerance. Although macrophages respond to the first exposure to 2-AA by enhancing ATP and acetyl-CoA production, long-term exposure to 2-AA leads to a tolerized state characterized by sustained reduced ATP and acetyl-CoA concentrations, a quiescent bioenergetic state with reduced mitochondrial pyruvate uptake and unresponsiveness to a second exposure.

Our results reveal that the decrease of ATP and acetyl-CoA in tolerized macrophages results from the 2-AA-mediated perturbation of the physical interaction between Esrra and Ppargc1a. The Ppargc1a/Esrra axis has not previously been implicated in tolerization but has been extensively studied in cancer, underscoring the novelty of our findings. Studies in cancer cells have shown that the physical interaction of human Esrra with Ppargc1a strongly enhances the binding of Esrra to DNA to induce the activation of the targeted genes (*Schreiber et al., 2003*), including *Mpc1,* responsible for the import of cytosolic pyruvate into mitochondria and PDH, that catalyzes the conversion of pyruvate to acetyl-CoA in mitochondria. Inhibition of Esrra has also been shown in cancer cell metabolism studies to interfere with pyruvate entry in mitochondria by inhibiting the expression of *Mpc1* (*Park et al., 2019*), and inhibition of Esrra led to decreased expression of Ppargc1a-controlled expression of mitochondrial genes in mice brains (*Bronte and Pittet, 2013*). Indeed, we find that the transcription of *Esrra, Ppargc1a,* and *Mpc1* genes and pyruvate uptake into mitochondria is reduced in tolerized macrophages. Given that pyruvate is a primary carbon source in the TCA cycle that fuels OXPHOS and the production of ATP into mitochondria, it explains the reduced ATP and acetyl-CoA levels observed in tolerized macrophages. Future studies will focus on unveiling the pathways by which macrophages respond to the bioenergetic changes identified and decipher their effect on the pyruvate cycle in host tolerance to infection (*Caslin et al., 2021*; *Lee, 2021*; *Li et al., 2022*).

We show that the unprecedented action of a QS bacterial small signaling molecule on the interaction between Esrra and Ppargc1a in the nucleus is hampered. Although it remains to be elucidated whether 2-AA interferes directly or indirectly with the Ppargc1a/Esrra axis, our data bring up the possibility that the reduced production of acetyl-CoA may be responsible since cholesterol is a known coactivator/ligand of Esrra (*Wei et al., 2016*), which is generated from acetyl-CoA (*Wei et al., 2016*). This possibility may also explain the reduced expression of *Esrra* and lower abundance since cholesterol is also required for its activation and autoregulation. However, it is also possible that 2-AA may antagonize the interaction between Esrra and Ppargc1a by binding to *Esrra*. Small molecules as antagonists of Esrra have been reported previously (*Chisamore et al., 2008*). The weaker binding of Esrra to the *Mpc1* promoter site, and consequently reduced *Mpc1* expression, explains the reduced pyruvate presence in mitochondria and that of ATP and mitochondrial quiescence. These findings open avenues for exploring novel therapeutic strategies through the overexpression of *Esrra* to counteract the tolerization induced by 2-AA and enhance clearance of *PA* infection. Interestingly, *Esrra* overexpression on breast cancer metastases promotes an efficient antitumor immune response selectively in the bone (*Bouchet et al., 2020*).

The observed cellular metabolic perturbances also align with our previous studies in skeletal muscle, which pointed to mitochondrial dysfunction triggered by 2-AA (*Tzika et al., 2013*; *Bandyopadhaya et al., 2016a*). Interestingly, the *PA* QS LasR-regulated homoserine lactone molecule, 3-oxo-C12-HSL, was reported to attenuate the expression of *Ppargc1a* and decrease the mitochondrial respiratory capacity in lung epithelial cells (*Maurice et al., 2019*). Moreover, another *PA* QS molecule, PQS, also regulated by LasR, was recently shown to induce organelle stress, including mitochondria, by disrupting the mitochondrial membrane potential in human macrophages (*Kushwaha et al., 2023*).

As opposed to 2-AA, which dampens the pro-inflammatory response, PQS increases pro-inflammatory cytokines (*Kushwaha et al., 2023*), consistent with the fact that PQS is an acute infection type molecule unrelated to chronic/persistent infections.

Previously, we reported that 2-AA tolerization induces histone deacetylation via HDAC1, reducing H3K18ac at the TNF-α promoter (*Bandyopadhaya et al., 2016b*). The findings with acetyl-CoA reduction, the primary substrate of histone acetylation, and the TNF-α transcription using UK5099 and ATP in 2-AA-treated macrophages are in support of the bioenergetics disturbances observed in macrophages and their link to epigenetic modifications we have shown to be promoted by 2-AA (*Bandyopadhaya et al., 2016b*). Macrophages exposed to 2-AA in the presence of exogenous ATP showed improved intracellular bacterial clearance and enhanced TNF-α levels, supporting the epigenetic interconnection of macrophages and cellular ATP responsiveness levels against infection. The exogenous addition of UK5099 that reverted the efficiency of the *mvfR* infected macrophages to clear the *PA* intracellular burden and the counteraction to the 2-AA effect by ATP addition that increases the clearance of *PA* intracellular burden support the importance of this energy-carrying molecule in *PA* persistence.

The results with the Mpc1 inhibitor, UK5099, suggest that the availability of pyruvate could underlie the mechanism of 2-AA-regulated HDAC/HAT-dependent control of transcription of pro-inflammatory mediators and bacterial clearance (*Pietrocola et al., 2015*). These results align with our previous findings, showing that establishing bacterial intracellular burden in macrophages is HDAC1 dependent (*Pietrocola et al., 2015*). It remains to be elucidated if HDAC/HAT-mediated histone modification directly regulates the expression of OXPHOS genes in 2-AA-tolerized macrophages, as it was shown that histone deacetylation downregulates OXPHOS in persistent *Mycobacterium tuberculosis* infection (*Chandran et al., 2015*; *Shi and Tu, 2015*). Given the complexity of 2-AA-mediated long-term effects, future omics studies combined with immune profiling may aid in deciphering the possible network of co-factors across different subpopulations of immune cells and the immunometabolic reprogramming related to 2-AA.

We have shown that although 2-AA tolerization leads to a remarkable increase in the survival rate of infected mice, it permits an HDAC1-dependent sustained presence of *PA* in mice tissues and intracellularly in macrophages (*Bandyopadhaya et al., 2012*; *Bandyopadhaya et al., 2016b*; *Bandyopadhaya et al., 2017*; *Chakraborty et al., 2023*). Here, we use 2-AA-producing and isogenic non-producing *PA* strains to confirm the 2-AA-mediated decrease in the central metabolite acetyl-CoA and the energy-carrying molecule ATP during infection. These murine infection studies provide strong evidence of the 2-AA biological relevance in reducing these metabolites in vivo. Taking together our previous and current in vivo findings provide compelling evidence that the metabolic alterations identified favor *PA* persistence in infected tissues.

That 2-AA permits *PA* to persist in infected tissues despite rescuing the survival of infected mice underscores its difference from that of LPS tolerization, which results in bacterial clearance (*Wheeler et al., 2008*) via the mechanism that relies on a set of different HDAC enzymes. LPS tolerization predominantly involves changes in H3K27 acetylation (*Lauterbach et al., 2019*), while 2-AA tolerization involves H3K18 modifications (*Bandyopadhaya et al., 2017*). The 2-AA-mediated effects reported here are also distinct from the epigenetic-metabolic reprogramming mediated by LPS (*Saeed et al., 2014*), which promotes endotoxin tolerance in immune cells by upregulating glycolysis and suppressing OXPHOS (*Liu et al., 2012*). Although 2-AA and LPS implicate different components and lead to different outcomes, both involve epigenetic mechanisms and immune memory.

Metabolic reprogramming upon infection may be pathogen-specific, with each pathogen impacting specific metabolic pathways that better fit its respective metabolic needs (*Galli and Saleh, 2020*). This was shown for infections of various tissues with *Chlamydia pneumonia*, *Legionella pneumophila*, *M. tuberculosis*, and *Salmonella* (*Ishida et al., 2014*; *Kunze et al., 2021*; *Shi et al., 2015*; *Pérez-Morales and Bustamante, 2021*) and, by our group, for *PA* infections (*Bandyopadhaya et al., 2012*; *Bandyopadhaya et al., 2016b*; *Tzika et al., 2013*). It would be interesting, however, to test whether 2-AA affects the killing efficacy of macrophages against other pathogens, as emerging evidence shows that synergistic or antagonistic interactions between clinically relevant microorganisms and host have important implications for polymicrobial infectious diseases (*Dhamgaye et al., 2016*).

While in this study, we focused on the role of Esrra mainly in pyruvate metabolism; future studies are needed to reveal other Esrra/Ppargc1a axis-dependent metabolic and immune pathways are

modulated by 2-AA. To this end, possible pathways to be interrgogated may be, the cholesterol synthesis pathway that relies on acetyl-CoA precursor, as cholesterol is a known coactivator/ligand of Esrra (*Wei et al., 2016*), fatty acid catabolism, which generates acetyl-CoA, and fatty acid oxidation (*Vega et al., 2000*).

This study unveils the unprecedented actions of a QS bacterial molecule in orchestrating cellular metabolic reprogramming in addition to the epigenetic reprogramming reported previously and has also been shown to promote a long-lasting presence of *PA* in the host (*Bandyopadhaya et al., 2012*; *Bandyopadhaya et al., 2016b*; *Chakraborty et al., 2023*). That 2-AA cross-tolerized macrophages to LPS corroborates our previous findings on the implication of epigenetic mechanism (*Bandyopadhaya et al., 2016b*) rather than ligand-receptor-mediated signaling-based mechanism and raises the possibility that this QS molecule may confer non-specific cross-protection. This is an aspect we plan to investigate in the future. The reported immunometabolic reprogramming contributes to a better understanding of the molecular and cellular mechanisms, biomarkers, and functional significance that may be involved in immune tolerance to persistent infection, providing for designing and developing innovative therapeutics and interventions. These approaches can focus on promoting host resilience against bacterial burden and safeguarding patients from recalcitrant persistent infections.

## Materials and methods

### Cell lines: source, authentication methods, and media used

RAW 264.7 and THP1 cells were obtained from ATCC. Short tandem repeat analysis (DNA finger-printing) was performed to determine the identity and uniqueness of a human line through Harvard Catalyst core facility. Periodic assays were performed to detect mycoplasma using PlasmoTest kit (InvivoGen). In addition, we used fluorescent Hoechst staining, Hoechst 33258 fluorescent dye, that binds specifically to DNA to reveal possible mycoplasma infections through their characteristic patterns of extracellular particulate or filamentous at ×500 magnification. BMDM cells were grown in Roswell Park Memorial Institute (RPMI) 1640 medium, while RAW 264.7 and THP1 cells were grown in Dulbecco's Modified Eagle Medium (DMEM).

### Conditions used for the exposures of the cells to 2-AA

Mouse BMDM cells were isolated from the femur of CD1 6-week-old male and female mice and used to estimate the levels of various metabolites (*Figure 2A*) following exposure and re-exposure to 2-AA (*Figure 1A*). The first exposure of naïve BMDMs ($10^6$/mL in six-well plates) was achieved with 400 µM 2-AA for 1, 6, or 48 hr. Cells exposed for 48 hr were used for the second 2-AA exposure. These cells were washed to remove residual 2-AA, left in the RPMI 1640 medium in the absence of 2-AA for either 24 or 106 hr, and re-exposed to 2-AA (200 µM) for 1 or 6 hr, respectively. For RAW 264.7 ($10^6$/mL in six-well plates) and THP-1 cells ($10^6$/mL in six-well plates) DMEM was used, the 2-AA concentration and conditions used were the same as those used for BMDM cells. The hours of the first round and second round of exposure to 2-AA are indicated in *Figure 1*.

For the LPS stimulation studies, cells were stimulated with 400 µM 2-AA or 100 ng/mL LPS for 6 hr and 800 µM of 2-AA for 48 hr. Cells receiving first exposure for 48 hr were washed and re-exposed to 400 µM of 2-AA or 100 ng/mL LPS for 6 hr.

### ATP and acetyl-CoA quantifications

The levels of ATP acetyl-CoA were assessed in BMDM and RAW 264.7 macrophage cells following exposure to 2-AA at the times indicated and by utilizing the ATP Assay kit (cholorimetric/fluorometric) (#ab83355, Abcam) and the PicoProbe AcCoA assay kit (ab87546, Abcam), respectively, according to the manufacturer's instruction. Quantifications of ATP and acetyl-CoA were performed in triplicates.

#### ATP

For ATP determination, macrophages were lysed and subsequently centrifuged at 14,000 rpm for 10 min at 4°C. The supernatants were transferred to a fresh Eppendorf tube. Standards and cell supernatants of 50 µL were added to a 96-well plate suitable for fluorescent analysis (black sides, clear bottom). A reaction mixture containing an ATP converter, probe, buffer, and developer mix was then added to all wells (50 µL) and incubated away from light at room temperature for 30 min. ATP

quantification was conducted fluorometrically at 535/587 nm using a plate reader and the following settings: $\lambda_{ex}$ = 535 nm; $\lambda_{em}$ = 587 nm. Fluorescence was measured using a microplate reader (Tecan Group Ltd, Männedorf, Switzerland).

### Acetyl-CoA

Acetyl-CoA content was assessed by first deproteinizing total cell fractions of macrophages using the perchloric acid and then centrifuging at 14,000 rpm for 10 min at 4°C. 50 µL of cells' supernatant sample CoASH were quenched to correct the background generated by free CoASH. Following the homogenization procedure described above, the samples were diluted with the reaction mix, and fluorescence was quantified using a plate reader and the settings as above with ATP.

## Seahorse assays

Seahorse analysis was performed according to the previously published protocols (*Van den Bossche et al., 2015*). Briefly, freshly prepared BMDM cells were reseeded in complete RPMI-1640 cells using Seahorse plates (Agilent cat. no. 103729100) at a density of $5 \times 10^4$ cells per well. Cells were exposed to 400 µM 2-AA for 1 or 24 hr (first exposure). Cells receiving first exposure for 24 hr were washed and re-exposed to 200 µM of 2-AA for 1 hr. Naïve cells were used as control. Prior to initiating Seahorse measurements, cells were washed, and Seahorse XF DMEM supplemented with 2 mM glutamine (Gibco cat. no. 25030-081) was added to each well. Cells were allowed to stabilize in a 37°C incubator without $CO_2$ for 1 hr. The Seahorse cartridge was hydrated and calibrated as per the manufacturer's instructions. The Mitochondrial Stress Test Kit (Agilent cat. no. 10395-100) was used (oligomycin at 1 µM, FCCP [1.5 µM], rotenone [0.5 µM], and antimycin A [0.5 µM]) with slight modifications according to the published protocol (*Van den Bossche et al., 2015*) that included injection of 25 mM glucose and sodium pyruvate (1 µM). All samples N=4 were run in a Seahorse XFe96 Analyzer, and data were analyzed using Wave and plotted using GraphPad Prism software.

## Isolation of cytosolic and mitochondrial fraction and pyruvate quantification

Pyruvate is produced in the cytosol and is transported into the mitochondria. RAW 264.7 cells were plated at $1 \times 10^6$ to incubate overnight at 37°C in a $CO_2$ incubator. Cells were exposed to 2-AA for 1, 3, 24, or 48 hr or re-exposed for 1 hr as described above. Mitochondria and cytosolic fractions of each group were isolated utilizing the Mitochondria Isolation Kit for cultured cells (#ab110170, Abcam) following the manufacturer's protocol. Briefly, cells were collected with a cell lifter and pelleted by centrifugation at 1000×$g$, frozen, and then thawed to weaken the cell membranes. The cells were resuspended in Reagent A and transferred into a pre-cooled Dounce Homogenizer. The homogenates were centrifuged at 1000×$g$ for 10 min at 4°C and saved as supernatant #1 for the cytosolic fractions. The pellet was resuspended in Reagent B, followed by repeat rupturing and centrifugation. The pellet was collected and resuspended in 500 µL of Reagent C supplemented with Protease Inhibitor cocktails (P8340, Sigma-Aldrich). Following separation, a Bradford assay was conducted to determine the protein concentration in each fraction. Pyruvate levels were determined in cellular fractions and mitochondrial fractions by using the Pyruvate Assay Kit (#ab65342, Abcam) according to the manufacturer's instructions. Briefly, after deproteinization using perchloric acid, the samples were neutralized in ice-cold 2 M KOH. 10 µL samples were incubated with reaction mix and kept on the plate at room temperature for 30 min in the dark. The absorbance was measured in a microplate reader (Tecan Group Ltd, Männedorf, Switzerland) at 570 nm, and the results were shown in three independent experiments.

## Bacterial strains and growth conditions

The *PA* strain known as Rif[R] human clinical isolate UCBPP-PA14 (also known as PA14) was used (*Rahme et al., 1995*). The bacteria were grown at 37°C in lysogeny broth (LB) under shaking and aeration or on LB agar plates containing appropriate antibiotics. PA14 and isogenic mutant Δ*mvfR* (*Cao et al., 2001*) cultures were grown in LB from a single colony to an optical density of 600 nm ($OD_{600}$) of 1.5, diluted 1:50,000,000 in fresh LB media, and grown overnight to an $OD_{600}$ of 3.0.

## Gentamicin protection assay

RAW 264.7 macrophage cells were plated on six-well cell culture-treated plates overnight in pyruvate-free DMEM. After 3 hr of incubation with 10 µM UK5099, 20 µM ATP or 400 µM 2-AA, cells were

infected with PA14 and isogenic mutant Δ*mvfR* at 5 MOI for 30 min at 37°C in 5% $CO_2$. Unbound bacteria were removed by washing once with cold DMEM. Afterward, the cells were incubated with 100 µg/mL of gentamicin for 30 min to eliminate residual extracellular bacteria. The cells were washed with DMEM, transferred to free medium without gentamicin and kept for 3 hr at 37°C in 5% $CO_2$. The infected cells were scraped after 3 hr, centrifuged at 500×$g$, and lysed in distilled water. The lysed cells were immediately diluted in PBS and plated on LB agar plates to assess bacterial presence. Bacterial colony-forming units (CFUs) were counted after incubating the plates overnight at 37°C. Untreated cells infected with PA14 were set as 100% and the reduction of the bacterial load was expressed as %CFU.

## Pharmacological inhibitors and ATP supplementations

For the OXPHOS inhibition assay, RAW 264.7 macrophage cells were treated with UK5099 (10 µM, Sigma-Aldrich, dissolved in DMSO) 3 hr prior to first or second rounds of 2-AA exposure. RAW 264.7 macrophages supplemented with ATP (20 µM, Sigma-Aldrich, dissolved in PBS) received ATP at the time of first and second 2-AA exposure.

## Co-IP assay

For all immunoprecipitation assays, protein A/G agarose Magnetic beads (Pierce) were used after washing in 1× IP buffer. Nuclear lysates from RAW 264.7 tolerized cells exposed to 2-AA for 24 hr or non-exposed cells were diluted in 1× IP buffer, and 500 µg of protein was taken for each experiment. The lysates were precleared by using unbound 50 µL protein A/G magnetic beads for 2 hr at room temperature on a rotator. Precleared lysates were either incubated with Esrra, Ppargc1a, or rabbit IgG antibody overnight at 4°C. 100 µL protein A/G magnetic beads were used to pull down the antibody-protein complex and washed twice with IP buffer to remove unbound proteins. Magnetic beads were then eluted in 2× Laemmli SDS-PAGE loading buffer. Immunoprecipitated Esrra and Ppargc1a was detected by western blot analyses, using conformational-specific Anti Rabbit antibody (TruBlot).

## Immunoblotting analysis

Cells were plated at 6×$10^5$ cells per well in six-well plates. Cells were washed with PBS and subsequently lysed using RIPA lysis buffer containing 1 mM phenylmethylsulfonyl fluoride. 20 µg of proteins were separated by electrophoresis on any KD (Kilo Dalton) (Bio-Rad, cat no. 4569033) SDS-polyacrylamide gel. Proteins were transferred to a 0.2 µm polyvinylidene fluoride membrane (Millipore, Billerica, MA, USA) using a Bio-Rad semi-dry instrument. After blocking with 5% BSA in TBS containing 0.1% Tween-20 for 1 hr at room temperature, the membranes were incubated with a primary antibody Esrra (Abcam, #ab76228), Mpc1 (D2L91, Cell Signaling), and anti-β-actin (cat no. sc-47778) (Santa Cruz Biotechnology) overnight at 4°C. Following washing, the membranes were incubated with an anti-rabbit secondary antibody, and the bands were detected by SuperSignal West Pico Chemilumi-nescent Substrate (Thermo Scientific) reaction, according to the manufacturer's instructions. The blots were visualized in the ChemiDOC Imaging system (Bio-Rad Laboratories, Inc, Hercules, CA, USA). The bands were analyzed densitometrically using QuantityOne software (Bio-Rad).

## ChIP and ChIP-qPCR

For ChIP studies, RAW 264.7 macrophage cells exposed to 2-AA for 24 hr were cross-linked in 1% (vol/vol) methanol-free formaldehyde for 10 min and then placed in 0.125 M glycine for 5 min at room temperature. Using the truChIP High Cell Chromatin Shearing kit (Covaris, USA), cells were prepared for sonication according to the Covaris protocol. Approximately $1 \times 10^7$ cells were plated in a 12 mm × 12 mm tube and subjected to shearing with the Covaris S220 sonicator for 8 min (140 peak power, 5 duty factor, 200 cycles/burst). The Magna ChIP A/G kit (Millipore, USA) was used for the subsequent immunoprecipitations according to the manufacturer's protocol. Briefly, chromatin from approximately $10^6$ cells was incubated overnight at 4°C with 2 µg of anti-Esrra or anti-Ppargc1a (Abcam, USA) ChIP-grade antibody and 20 µL of A/G magnetic beads. The beads were washed seri-ally (5 min each) with low-salt wash buffer, high-salt wash buffer, LiCl wash buffer, and TE buffer from the kit at 4°C. Chromatin was eluted with elution buffer containing Proteinase K at 62°C for 4 hr, then incubated at 95°C for 10 min. DNA was isolated by column purification (QIAquick PCR purification kit).

Real-time ChIP-qPCR was performed with the Brilliant II SYBR green super mix (Agilent, USA). Forward (AGTGGTGACCTTGAACTTCCC) and reverse (CTGAAGACGACCTTCCCCTT) primers were chosen to amplify a genomic locus of *Mpc1* promoter, which had a putative ERR-binding site (TNAAG-GTCA) at 1582 bp upstream of the start site. Normalized values were calculated using the percent-input method relative to the IgG. The assay was performed three times.

## RNA extraction and RT-qPCR

Total RNA from all the groups mentioned above was isolated from approximately $2 \times 10^6$ cells with the RNeasy minikit (QIAGEN, USA), and cDNA was prepared with the iScript Reverse transcription kit (Bio-Rad, USA), as per the manufacturer's instruction. Real-time PCR was conducted using the PowerUP SYBR Green Master mix (Applied Biosystems, USA) and primer sets for mouse *Esrra* (forward: ACTACGGTGTGGCATCCTGTGA; reverse: GGTGATCTCACACTCATTGGAGG), *Ppargc1a* (forward: GAATCAAGCCACTACAGACACCG; reverse: CATCCCTCTTGAGCCTTTCGTG), MPC-1 (forward: CTCCAGAGATTATCAGTGGGCG; reverse: GAGCTACTTCGTTTGTTACATGGC), TNF-α (forward: GGTGCCTATGTCTCAGCCTCTT; reverse: GCCATAGAACTGATGAGAGGGAG) and mouse 18S rRNA (forward: GTTCCGACCATAAACGATGCC; reverse: TGGTGGTGCCCTTCCGTCAAT). The transcript levels of all the genes were normalized to 18S rRNA with the ΔΔCT method. The relative expression was calculated by normalizing transcript levels to those of PA14-infected cells. The assay was conducted in triplicate; means and standard deviations were calculated for each group.

## Animal infection and metabolites assessment experiments

The full-thickness thermal burn injury and infection (BI) model (*Bandyopadhaya et al., 2012*; *Bandyopadhaya et al., 2016b*; *Maura et al., 2018*) was used to assess the effect of 2-AA on metabolic alterations in the spleens of 6-week-old CD1 male mice and bacterial burden (Charles River Labs, USA). A full-thickness thermal burn injury involving 5–8% of the total body surface area was produced on the shaved mouse abdomen dermis, and an inoculum of ~$1 \times 10^4$ PA14 and Δ*mvfR* (*Cao et al., 2001*) cells in 100 µL of MgSO$_4$ (10 mM) was injected intradermally into the burn eschar. For the groups that received 2-AA, mice were injected intraperitoneally with 100 µL of 2-AA (6.75 mg/kg). The entire procedure was done under the influence of anesthesia. One of the groups of mice infected with the PA14 isogenic mutant Δ*mvfR* (*Cao et al., 2001*) also received 100 µL of 2-AA in PBS (6.75 mg/kg) at the time of infection (Δ*mvfR* + 2-AA) and served as an additional control. CFU counts from rectus abdominus muscle (underlying the burn-infected tissue) were assessed in groups of four mice each at 1, 5, and 10 days post-BI by plating diluted muscle homogenate on Pseudomonas Isolation Agar (Sigma-Aldrich) plates containing rifampicin (50 mg/L). Spleen samples from mice were collected for metabolite analyses from all mice groups at 1, 5, and 10 days post-BI to assess ATP (#ab83355 Abcam) and acetyl-CoA (#ab87546, Abcam) as described above. Tissues were homogenized, 100 mg of homogenate was centrifuged at 4°C for 15 min at 10,000×*g*, and supernatants were collected. The supernatant was mixed with 400 µL of 1 M perchloric acid. The deproteinized supernatant was neutralized by 3 M KHCO$_3$. The ATP and acetyl-CoA assays were performed as described in the in vitro section.

## Electron microscopy studies

RAW 264.7 macrophages exposed to 2-AA and corresponding controls were fixed with 2% glutaraldehyde in 0.1 M cacodylate buffer and post-fixed in 1% OsO$_4$ in 0.1 M cacodylate buffer for 1 hr on ice. The cells were stained all at once with 2% uranyl acetate for 1 hr on ice, after which they were dehydrated in a graded series of ethanol (50–100%) while remaining on ice. Ultrathin (70 nm) sections were cut using a Leica EMUC7 ultramicrotome and collected onto formvar-coated grids (EMS, Hatfield, PA, USA). Sections were contrast-stained using 2.0% aqueous uranyl acetate. Grids were examined at 80 kV in a JEOL 1011 transmission electron microscope (Peabody, MA, USA) equipped with an AMT digital camera and proprietary image capture software (Advanced Microscopy Techniques, Danvers, MA, USA).

## Acknowledgements

We thank Dr. Vamsi Mootha and Sneha Rath for access and help with using the Seahorse instrument, respectively. We also thank Dr. Diane Capen and Dr. Dennis Brown for the guidance and processing

of the ultramicroscopy images. This work was supported by the NIH award R01AI134857, The John Lawrence Massachusetts General Hospital Research Scholar Award and Shriner's grant 83009 to LGR, and the Shriner's grant 85132 to AAT. The funders had no role in the study design, data collection, analysis, decision to publish, or manuscript preparation.

## Additional information

### Competing interests

Laurence G Rahme: has a financial interest in Spero Therapeutics, a company developing therapies to treat bacterial infections. L.G.R.'s financial interests are reviewed and managed by Massachusetts General Hospital and Partners Health Care in accordance with their conflict-of-interest policies. No funding was received from Spero Therapeutics, and it had no role in study design, data collection, analysis, interpretation, or the decision to submit the work for publication. The other authors declare that no competing interests exist.

### Funding

| Funder | Grant reference number | Author |
|---|---|---|
| National Institute of Allergy and Infectious Diseases | R01AI134857 | Arijit Chakraborty<br>Arunava Bandyopadhaya<br>Vijay K Singh<br>Filip Kovacic<br>Sujin Cha<br>William M Oldham<br>A Aria Tzika<br>Laurence G Rahme |
| Shriners Hospitals for Children | 83009 | Arijit Chakraborty<br>Laurence G Rahme |
| Shriners Hospitals for Children | 85132 | Vijay K Singh<br>A Aria Tzika<br>Laurence G Rahme |

The funders had no role in study design, data collection and interpretation, or the decision to submit the work for publication.

### Author contributions

Arijit Chakraborty, Data curation, Formal analysis, Validation, Investigation, Visualization, Methodology, Writing – original draft; Arunava Bandyopadhaya, Conceptualization, Formal analysis, Investigation, Visualization, Methodology; Vijay K Singh, Investigation, Visualization, Methodology; Filip Kovacic, Data curation, Visualization, Writing – original draft, Writing – review and editing; Sujin Cha, Investigation; William M Oldham, Resources, Data curation, Writing – original draft; A Aria Tzika, Resources; Laurence G Rahme, Conceptualization, Resources, Data curation, Formal analysis, Supervision, Funding acquisition, Validation, Writing – original draft, Project administration, Writing – review and editing

### Author ORCIDs

Arijit Chakraborty ⓘ https://orcid.org/0000-0002-0883-6385
Vijay K Singh ⓘ https://orcid.org/0000-0003-4872-4545
Filip Kovacic ⓘ https://orcid.org/0000-0002-0313-427X
Laurence G Rahme ⓘ https://orcid.org/0000-0002-5374-4332

### Ethics

All animals were handled according to the approved protocol by the Institutional Animal-Care and Use Committee (IACUC) of Massachusetts General Hospital (protocol no. 2006N000093). No randomization or exclusion of data points was applied. The study was performed in strict accordance with the recommendations in the Guide for the Care and Use of Laboratory Animals of the National Institutes of Health.

Reviewer #1 (Public Review): https://doi.org/10.7554/eLife.97568.3.sa1
Reviewer #2 (Public Review): https://doi.org/10.7554/eLife.97568.3.sa2
Author response https://doi.org/10.7554/eLife.97568.3.sa3

## Additional files

### Supplementary files
• MDAR checklist

### Data availability
All data generated or analysed during this study are included in the manuscript and supporting files.

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
