## [Editor Report · eLife assessment]

This **important** study demonstrates that the *Pseudomonas aeruginosa*-derived quorum sensing signal, 2-aminoacetophenone, induces immune tolerization in macrophages by perturbing metabolism, particularly in the context of mitochondrial respiration and bioenergetics. The authors present **convincing** evidence for 2-aminoacetophenone-mediated reduction of pyruvate transport into mitochondria, with downstream effects that result in reduced ATP production in tolerized macrophages. The work will be of interest to those studying host-pathogen interactions.

---

## [Referee Report · Reviewer #1 (Public Review)]

Their findings elucidate the mechanisms underlying 2-AA-mediated reduction of pyruvate transport into mitochondria, which impairs the interaction between ERRα and PGC1α, consequently suppressing MPC1 expression and reducing ATP production in tolerized macrophages.

This paper presents a novel discovery regarding the mechanisms through which PA regulates the bioenergetics of tolerized macrophages. This paper will provide valuable insights for the journal's broad readership of scientists.

---

## [Referee Report · Reviewer #2 (Public Review)]

Summary:

The study tries to connect energy metabolism with immune tolerance during bacterial infection. The mechanism details the role of pyruvate transporter expression via ERRalpha-PGC1 axis, resulting in pro-inflammatory TNF alpha signalling responsible for acquired infection tolerance.

Strengths:

Overall, the study is an excellent addition to the role of energy metabolism during bacterial infection. The mechanism-based approach in dissecting the roles of metabolic coactivator, transcription factor, mitochondrial transporter and pro-inflammatory cytokine during acquired tolerance towards infections indicates a detailed and well-written study. The in vivo studies in mice nicely corroborate with the cell line-based data, indicating the requirement for further studies in human infections with another bacterial model system.

Weakness:

Revised version doesn't have much weakness as authors have performed some of the critical experiments to answer the concerns. Moreover, authors promted that a few concerns like public data sets, etc are out of scope of this work or they will perform such experiments in future.

---

## [Author Response]

The following is the authors’ response to the original reviews.

**Public Reviews:**

**Reviewer #1 (Public Review):**

We thank the reviewer for the time and effort in reviewing our revised manuscript and are grateful for their constructive comments and for acknowledging the significance of our work.

Summary:Their findings elucidate the mechanisms underlying 2-AA-mediated reduction of pyruvate transport into mitochondria, which impairs the interaction between ERRα and PGC1α, consequently suppressing MPC1 expression and reducing ATP production in tolerized macrophages. While the data presented is intriguing and the paper is well-written, there are several points that warrant consideration. The authors should enhance the clarity, relevance, and impact of their study.Strengths:This paper presents a novel discovery regarding the mechanisms through which PA regulates the bioenergetics of tolerized macrophages.Weaknesses:The relevance of the in vivo model to support the conclusions is questionable. Further clarification is needed on this point.

We appreciate the reviewer’s comment. Our conclusion that 2-AA decreases bioenergetics while sustains bacterial burden is further supported by additional in vivo data we present now in Fig. S5. To strengthen the relevance of our in vivo data, we performed additional in vivo experiments. In this set of in vivo studies, mice received the first exposure to 2-AA by injecting 2-AA only and the 2nd exposure through infection with PA14 or ΔmvfR four days post-2-AA injection. As shown in the supplementary Figure S5 the levels of ATP and acetyl-CoA in the spleen of infected animals and the enumeration of the bacterial counts were the similar between PA14 or ΔmvfR receiving the 1st 2-AA exposure and agree with the “one-shot infection” findings presented in Figure 5 with the PA14 or ΔmvfR+2-AA infected mice or those receiving 2-AA only. These results are consistent with our previous findings showing that 2-AA impedes the clearance of PA14 (Bandyopadhaya et al. 2012; Bandyopadhaya et al. 2016; Tzika et al. 2013) and provide compelling evidence that the metabolic alterations identified may favor PA persistence in infected tissues.

**Reviewer #2 (Public Review):**

We thank the reviewer for the time and effort in reviewing our revised manuscript and are grateful for their constructive comments and for acknowledging the significance of our work.

Summary:The study tries to connect energy metabolism with immune tolerance during bacterial infection. The mechanism details the role of pyruvate transporter expression via ERRalpha-PGC1 axis, resulting in pro-inflammatory TNF alpha signalling responsible for acquired infection tolerance.Strengths:Overall, the study is an excellent addition to the role of energy metabolism during bacterial infection. The mechanism-based approach in dissecting the roles of metabolic coactivator, transcription factor, mitochondrial transporter, and pro-inflammatory cytokine during acquired tolerance towards infections indicates a detailed and well-written study. The in vivo studies in mice nicely corroborate with the cell line-based data, indicating the requirement for further studies in human infections with another bacterial model system.Weaknesses:The authors have involved various mechanisms to justify their findings. However, they have missed out on certain aspects which connect the mechanism throughout the paper. For example, they measured ATP and acetyl COA production linked with bacterial re-exposures and added various targets like MCP1, EER alpha, PGC1 alpha, and TNF alpha. However, they skipped PGC1 alpha levels, ATP and acetyl COA in various parts of the paper. Including the details would make the work more comprehensive.

We appreciate the reviewer’s comments and apologize for omitting the PGC-1α levels. Per the reviewer’s suggestion, we have added the PGC-1α transcript levels (Figure 4C) in the section describing 2-AA-mediated dysregulation of the ERRα and MPC1 transcription (lines 243-252). Moreover, we have added Figure S5, which shows additional ATP and acetyl CoA levels in vivo. In our view, ATP and acetyl-CoA levels are shown in all appropriate settings, interrogating the bioenergetics, including in the presence of bacteria and in their absence, where only 2-AA is added. Please see Figures 1 and 5 and the newly added Figure S5.

The use of public data sets to support their claim on immune tolerance is missing. Including various data sets of similar studies will strengthen the findings independently.

Suppose we understand correctly the reviewer’s comment regarding public data sets on immune tolerance. In that case, we are referring to our data since there are no published data from other groups on 2-AA tolerization and because the outcome of the 2-AA effect on the bacterial burden differs from that of LPS. Therefore, this study did not consider comparing with published data from LPS.

**Reviewer #1 (Recommendations For The Authors):**
(1) Animal model: The authors appropriately initiated the study with an in vitro tolerization model involving 2-AA re-exposure, providing foundational insights for further investigation. However, the rationale for the one-shot injection in the in vivo model lacks clarity. To strengthen the relevance of the in vivo data, the authors should consider establishing a model involving bacterial re-exposure, such as a two-challenge paradigm with antibiotic treatment in between. This approach would allow for the examination of peritoneal macrophages harvested from mice, assessing ATP levels, acetyl CoA, TNF production, and bacterial counts. Such an approach would better align the in vivo findings with the in vitro experiments, confirming the role of tolerized macrophages in controlling PA infection in the presence of 2-AA.

We thank the reviewer for this comment. Indeed, we have performed a similar two-challenge paradigm study in which first exposure to 2-AA is achieved by injecting 2-AA, and 2nd exposure through infection with PA14 or ΔmvfR four days post -2-AA injection. The results of Figure S5 can be directly compared with those in Fig 5 in vivo studies. As shown in supplementary Figure S5 the levels of ATP and acetyl-CoA in the spleen of infected animals and the enumeration of the bacterial counts agree with the “one-shot infection” presented in Fig 5 (PA14 or ΔmvfR+2-AA). Figure S5 study although not included initially to simplify data presentation, it was performed in parallel with Fig 5 and thus they can be directly compared.

(2) Exogenous ATP treatment: It is crucial to explore whether 2-AA re-exposure suppresses inflammasome activation and whether this suppression can be reversed by exogenous ATP treatment. Specifically, the authors should investigate whether NLRP3 inflammasome activation is inhibited in tolerized macrophages and whether such activation is necessary for host defense. Clarifying these points would provide valuable insights into the mechanisms underlying macrophage tolerization induced by 2-AA.

Excellent point. We agree, indeed, this is planned in the near future.

(3) Figures 4C and D: The authors should exercise care in describing these figures. For instance, line 263 states that "UK5099 had no effect on the PA14 burden in macrophages," which requires correction for accuracy.

We apologize and rephrase this sentence and other sentences referring to Fig 4D and 4E in this section. Please see the highlighted sentences in the results section referring to Fig 4. For example, “The addition of the UK5099 inhibitor strongly enhanced the bacterial intracellular burden in ΔmvfR infected macrophages compared to the non-inhibited ΔmvfR infected cells, reaching a similar burden to those infected with PA14 (Fig. 4D)”.

(4) ERRα expression: While the study intriguingly demonstrates a decrease in ERRα levels in tolerized macrophages following exposure to 2-AA, the discussion of this finding is lacking. It is worth exploring the possibility of increasing ERRα expression to counteract the tolerization induced by 2-AA and enhance clearance of PA infection. This avenue should be thoroughly discussed in the manuscript's Discussion section, offering insights into potential therapeutic strategies to mitigate the effects of 2-AA on macrophage function.

Thank you so much for this additional comment. We have now included this point in the discussion section (lines 373-376).

**Reviewer #2 (Recommendations For The Authors):**
Overall, the study is an excellent addition to the role of energy metabolism during bacterial infection. The mechanism-based approach in dissecting the roles of metabolic coactivator, transcription factor, mitochondrial transporter, and pro-inflammatory cytokine during acquired tolerance indicates a detailed and well-written study. However, connecting the mechanisms often was not reflected in some of the experiments, and answering a few concerns/suggestions will undoubtedly improve the study's readability, appeal, and overall impact on a broader audience.(1) The authors should rephrase the title if possible. The title indicates 2AA as a bacterial quorum sensing signal; however, throughout the manuscript, there are no studies associated with actual quorum sensing in bacteria.

Thank you for this comment. However, the title indicates 2-AA as a quorum sensing molecule because the synthesis of this signaling molecule is uniquely regulated by quorum sensing. Because of its importance in the virulence of *Pseudomonas aeruginosa* and its regulation by quorum sensing, we feel that it is appropriate to refer to it as such.

(2) The authors generalised immunotolerance and memory of 2AA-exposed cells to broad-spectrum microbial exposure by just testing with LPS exposure. I would suggest they test at least 2 more heterologous microbial products known to illicit response and confirm their claim from Figure 1.

We appreciate the reviewer’s comment. We intend not to generalize immunotolerance and memory of 2-AA exposed cells to broad-spectrum microbial exposure. Moreover, since the manuscript is not focused on comparing other bacterial molecules to 2-AA and multiple studies have focused on LPS tolerance, we tested LPS only in the manuscript.

(3) LPS triggers ATP production through glycolysis in nitric oxide (NO) dependent mechanisms in various immune and non-immune cells. The authors should study the concentrations of NO, Glucose, and Pyruvate levels to clarify the mechanism of energy dynamics and the source of ATP and Acetyl CoA generated/scavenged during primary and secondary exposures to both 2AA and LPS.

We agree that a cross-tolerization experiment using 2-AA and LPS would reveal interesting insights into immune response during PA infections. However, this is out of the scope of this article. Please notice that the mechanism of 2-AA and LPS tolerization is mechanistically distinct, e.g. they rely on different HDAC enzymes, and LPS tolerization predominantly involves changes in H3K27 acetylation (Lauterbach et al. 2019). In contrast, 2-AA tolerization involves H3K18 modifications (Bandyopadhaya, Tsurumi, and Rahme 2017). For this reason, the complexity of such interactions would require a comprehensive set of experiments that are not part of the focus of this study.

(4) Immunogenic triggers often rapidly alter mitochondrial membrane potential, which alters oxygen consumption rates. However, the authors tend to generalize energy homeostasis and claim the deregulation of OXPHOS-inducing quiescent phenotype depending upon OCR measurements from Figure 1D. The authors must evaluate mitochondrial health and membrane potential during first and second exposure in a time-dependent manner to strengthen their theory of mitochondrial dysfunction. The authors should also check the phenomena in vivo (mice exposed to infection) if possible.

Thank you for this suggestion. We now include electron microscopy images of mitochondria isolated from macrophages exposed to 2-AA. Results revealed that 2-AA alters mitochondrial morphology and cristae, supporting the mitochondrial dysfunctionality caused by 2-AA. These results are shown in Figure S4 and lines 185-188.

(5) Since both MCP1 and MCP2 transporters are known to transport pyruvate to mitochondria, checking both MCP1 and 2 at transcript and protein levels in exposed cells will be essential. I suggest authors use MCP inhibitors or use RNA interference against MCPs to check the effect on tolerance of the cells exposed for a second time.

To our understanding, mitochondrial pyruvate carrier proteins, MPC1 and MPC2, form a hetero-oligomeric complex in the inner mitochondrial membrane to facilitate pyruvate import into mitochondria (McCommis and Finck 2015). We also used UK5099 an MPC carrier inhibitor for enumeration of bacterial load in macrophages in Figure 4 and observed a similar effect as 2-AA suggesting a similar mechanism of action.

(6) The pyruvate levels of mitochondria in Figure 2A are shallow, and the authors claim statistical significance within a 1.5-fold change. The authors should cross-check the number of mitochondria they are isolating while estimating pyruvate from only mitochondrial fractions. Another point is, correlating mitochondrial pyruvate with the burst of ATP during first exposure in comparison to second exposure, one can argue that the number of mitochondria is variable between the exposures leading to a change in pyruvate amount (mitochondria number increases to compensate for the first exposure and decreases quickly to maintain homeostasis and remains quiescent during a second exposure due to activation of compensatory immune mechanism towards primary exposure). How do authors address the issue?

Our electron microscopic studies indicate that although after 2-AA exposure, no reduction in mitochondrial numbers is observed in macrophages, alterations in mitochondrial morphology and cristae are observed. Please also see our answer to point # 4.

(7) The authors claim that ERR alpha regulates MCP1 transcription via activation of ERRalpha-PGC1 alpha axis and tolerization in cells to second exposure is due to impairment of the axis (Figure 3). PGC1 alpha is known to be induced during various metabolic, physiological, and immune-challenge-related stress in a tissue-dependent manner. In this context, one should expect changes in transcript and protein levels of PGC1 alpha. The authors must study PGC1 alpha levels with time-dependent exposures. LPS was shown to induce oscillations in PGC1 alpha levels in a tissue-specific manner. In experiments, authors should verify if such oscillations persist during time-dependent exposure, emphasising mitochondrial uncoupling that might get dampened during re-exposures to microbial challenges.

We appreciate the suggestion. We have now included PGC-1α (Figure 4C) transcript levels, which show the same profile as the transcript levels of ERRα and MPC1. Please note that PGC-1α is only one of several ERRα co-activators; therefore, the amount of ERRα protein is the most relevant assessment regarding the activation of the MPC1 transcription.

(8) The authors claim that ERRalpha induces MCP1 through ChIP data in Figure 3. However, the physical verifications at mRNA levels and mutational/inhibitor-based experiments are missing. The authors should study the alterations of MCP1 mRNA in relation to exposures and inhibitors of ERRalpha and PGC1 alpha to strengthen their work.

This is an interesting approach; however, this experiment exceeds the scope of our manuscript. We will certainly consider this suggestion in our future experiments. Thank you.

(9) Publicly available data sets with LPS exposures should be analyzed for gene sets pertaining to mitochondrial OXPHOS, metabolism, immune response, etc. This will support the authors' work and provide a global overview of transcriptome associated with immune tolerance.

We appreciate the reviewer’s comment. For the reasons explained in #3 point and because the bacterial burden outcome of the 2-AA effect is different from that of LPS, comparison with LPS published data was not considered in this study. We agree that in the future, a comprehensive comparison of whole genome transcriptome studies between LPS and 2-AA may reveal important insights that may also help better understand and potentially classify the immune tolerance triggered by 2-AA.

(10) In Figure 4, the authors study the role of MCP1 and associated pyruvate-dependent bacterial clearance during tolerization and associate them with a decrease in TNF alpha. I would suggest the addition of an ERR alpha inhibitor in these experiments. It is not clear as to why (mechanism) TNF alpha transcription was affected via pyruvate transport during bacterial exposure. I would suggest that the authors clarify the mechanism of TNF alpha activation/inactivation and its association with energy metabolism during acquired tolerance.

This is an excellent suggestion, given that a similar effect of ERRα on TNF-α was observed by other researchers (Chaltel-Lima et al. 2023). Here, to clarify the mechanism of TNF alpha activation/inactivation and its association with energy metabolism, we elaborate on this aspect in the discussion section.

Lines 388-393. The text reads:

Previously, we reported that 2-AA tolerization induces histone deacetylation via HDAC1, reducing H3K18ac at the TNF-α promoter (Bandyopadhaya et al. 2016). The findings with acetyl-CoA reduction, the primary substrate of histone acetylation, and the TNF-α transcription using UK5099 and ATP in 2-AA treated macrophages are in support of the bioenergetics disturbances observed in macrophages and their link to epigenetic modifications we have shown to be promoted by 2-AA (Bandyopadhaya et al. 2016)

(11) It is surprising that authors specifically target TNF alpha as a pro-inflammatory cytokine during tolerance. Various reports of cytokines and immune modulatory factors play a vital role in immune tolerance upon bacterial exposure. I would suggest authors perform cytokine profiling or check public data sets to specify their reason for choosing TNF alpha.

The choice of TNF-α is based on the results obtained in our previous study (Bandyopadhaya et al. 2016).

Bandyopadhaya, A., M. Kesarwani, Y. A. Que, J. He, K. Padfield, R. Tompkins, and L. G. Rahme. 2012. 'The quorum sensing volatile molecule 2-amino acetophenon modulates host immune responses in a manner that promotes life with unwanted guests', PLoS pathogens, 8: e1003024.

Bandyopadhaya, A., A. Tsurumi, D. Maura, K. L. Jeffrey, and L. G. Rahme. 2016. 'A quorum-sensing signal promotes host tolerance training through HDAC1-mediated epigenetic reprogramming', Nat Microbiol, 1: 16174.

Bandyopadhaya, A., A. Tsurumi, and L. G. Rahme. 2017. 'NF-kappaBp50 and HDAC1 Interaction Is Implicated in the Host Tolerance to Infection Mediated by the Bacterial Quorum Sensing Signal 2-Aminoacetophenone', Front Microbiol, 8: 1211.

Chaltel-Lima, L., F. Domínguez, L. Domínguez-Ramírez, and P. Cortes-Hernandez. 2023. 'The Role of the Estrogen-Related Receptor Alpha (ERRa) in Hypoxia and Its Implications for Cancer Metabolism', Int J Mol Sci, 24.

Lauterbach, M. A., J. E. Hanke, M. Serefidou, M. S. J. Mangan, C. C. Kolbe, T. Hess, M. Rothe, R. Kaiser, F. Hoss, J. Gehlen, G. Engels, M. Kreutzenbeck, S. V. Schmidt, A. Christ, A. Imhof, K. Hiller, and E. Latz. 2019. 'Toll-like Receptor Signaling Rewires Macrophage Metabolism and Promotes Histone Acetylation via ATP-Citrate Lyase', Immunity, 51: 997-1011 e7.

McCommis, K. S., and B. N. Finck. 2015. 'Mitochondrial pyruvate transport: a historical perspective and future research directions', Biochem J, 466: 443-54.

Tzika, A. A., C. Constantinou, A. Bandyopadhaya, N. Psychogios, S. Lee, M. Mindrinos, J. A. Martyn, R. G. Tompkins, and L. G. Rahme. 2013. 'A small volatile bacterial molecule triggers mitochondrial dysfunction in murine skeletal muscle', PloS one, 8: e74528.